# Deep Coupling Learning for Solving PDEs

Lingshi Meng [1]   Haosen Shi [1]   Sinno Jialin Pan [1]

## Abstract

Physics-Informed Neural Networks (PINNs) represent a significant advancement in computational methods for solving partial differential equations (PDEs). However, adopting deeper neural network architectures in PINNs presents significant challenges, as derivative-related computations over the input can lead to numerical instabilities. These complications extend beyond traditional vanishing and exploding gradients to include vanishing and exploding differentials, with both phenomena becoming more severe as networks grow deeper. By examining the computation graph of derivatives in deep neural networks, we identify key bottlenecks behind these instabilities and introduce Coupling Layers with carefully regulated spectral norms of Jacobian matrices to stabilize deep PINN training. Comprehensive evaluations show that our approach surpasses conventional shallow PINN methods and alternative deep PINN designs across a range of challenging problems, particularly in cases featuring high-frequency solution components.

## 1. Introduction

Neural networks have revolutionized the field of scientific computing by offering a data-driven approach to solving partial differential equations (PDEs) that govern physical systems. Unlike traditional numerical methods such as finite element analysis (FEA) (Belytschko et al., 2014) and finite difference methods (FDM) (LeVeque, 2007), which often require domain discretization and become computationally expensive for high-dimensional problems. In contrast, neural networks provide a scalable and mesh-free alternative (Lagaris et al., 1998; Han et al., 2018). Physics-Informed Neural Networks(PINNs) (Raissi et al., 2019; Karniadakis et al., 2021) have emerged as a promising frame-work, embedding physical laws directly into the loss function of neural networks. This integration enables PINNs to tackle both forward and inverse problems without extensive labeled datasets, making them particularly effective across diverse applications, including fluid dynamics (Mao et al., 2020), heat transfer (Cai et al., 2021), and quantum mechanics (Raissi et al., 2019). When compared to traditional numerical methods, PINNs demonstrate notable strengths in managing irregular geometries, scaling to high-dimensional problems, and naturally incorporating observational data (Wang et al., 2021a). Nevertheless, despite their promising capabilities, PINNs continue to struggle with achieving comparable accuracy and reliability to established numerical methods, especially when confronted with complex geometrical structures or solutions with high-frequency components that require fine-scale resolution (Sirignano & Spiliopoulos, 2018; E & Yu, 2018).

Specifically, high-frequency problems pose a particular difficulty for neural networks due to their limited ability to capture fine-scale features (Tancik et al., 2020; Krishnapriyan et al., 2021). While traditional numerical methods can handle such problems by refining the computational mesh, PINNs often struggle to converge to accurate solutions without specialized architectural modifications (Raissi et al., 2019; Wang et al., 2021a). This limitation has led to a growing interest in developing techniques to enhance the ability of PINNs to model high-frequency phenomena, which is critical for their adoption in applications (Karniadakis et al., 2021; Sirignano & Spiliopoulos, 2018).

Current approaches to addressing high-frequency problems in PINNs predominantly rely on frequency-aware components. For instance, Fourier feature embeddings (Tancik et al., 2020) and sinusoidal activation functions (Sitzmann et al., 2020) have been shown to improve the performance of PINNs on high-frequency tasks. While these training methods are effective for high-frequency PDEs, they often require careful tuning of hyperparameters, such as the frequency range or the scale of the embedding, which can limit their generalization (Wang et al., 2022c; Glorot & Bengio, 2010). These approaches do not fully exploit the potential of deep architectures, which offer great flexibility in expressing functions with complex frequency in Fourier space.

Here, we hypothesize that a sufficiently deep PINN can

---

[1]Department of Computer Science and Engineering, The Chinese University of Hong Kong, Hong Kong SAR, China. Correspondence to: Sinno Jialin Pan <sinnopan@cse.cuhk.edu.hk>.

*Proceedings of the 43rd International Conference on Machine Learning*, Seoul, South Korea. PMLR 306, 2026. Copyright 2026 by the author(s).

accurately solve high-frequency, challenging PDE problems. Though this hypothesis is empirically supported by our experiments (see Section 4), training deeper PINNs is not straightforward. Unlike conventional neural networks, which optimize a loss function based solely on model outputs, PINNs must minimize a composite loss that incorporates both the direct output and partial derivatives derived from the governing equations. This dual objective makes PINNs particularly susceptible to gradient-related issues, such as vanishing gradients, during training (Glorot & Bengio, 2010; He et al., 2016). For instance, backpropagating first-order derivatives through a stacked Multilayer Perceptron (MLP) in PINNs involves repeated multiplication of layer-wise Jacobian matrices, which can result in exponentially diminishing matrix elements and impede training convergence (Wang et al., 2021a; Lu et al., 2021).

While ResNets (He et al., 2016) have been widely used in deep learning to alleviate vanishing gradients via skip connections, their application to PINNs has yielded inconsistent outcomes. Wang et al. (2024a) found that ResNet-based PINNs tend to accumulate errors as network depth increases, especially in problems demanding high precision (Wang et al., 2024a). Thus, there is a clear need for alternative architectural innovations that enable deeper PINN models to achieve high accuracy.

In this work, we introduce a novel network architecture that preserves coherent derivation structures, enabling the training of deep Physics-Informed Neural Networks (PINNs) with high precision. At its core is a **Coupling Block** framework, a concept adapted from architectural designs in normalizing flows (Dinh et al., 2015; 2017). Our **key contributions** are summarized as follows:[1]

1. **Empirical Validation of Depth in PINNs**: Through extensive experiments, we demonstrate that increasing the network depth of PINNs is critical to solving complex physical PDE problems. Deeper architectures offer the expressive capacity required to capture intricate solution features that shallow PINNs fail to represent accurately.

2. **Coupling Layers for Stable Differentiation**: We introduce a novel approach that ensures stable differentiation in deep PINNs via spectral norm regularization of Jacobian matrices and dynamic scaling control in the output layer. As demonstrated in Figure 2, our CoupledNet architecture effectively mitigates both vanishing gradients, and gradient explosion issues typical of ResNet-based implementations.

3. **Solving PDEs with Diverse Characteristics**: Unlike existing methods that require problem-specific tuning,

such as Fourier features for high-frequency PDEs, our model generalizes robustly across PDEs with varying characteristics. It achieves this without manual hyperparameter adjustments, offering a versatile and practical solution for a broad range of physical systems.

The CoupledNet design introduces additional operations within each coupling block, which increases the wall-clock time per training step in practice. However, this overhead is accompanied by faster convergence. In addition, while CoupledNet may not always yield the best score on relatively simple PDE benchmarks, its architectural inductive bias preserves competitive accuracy in these cases and provides substantially larger gains on more difficult PDE systems.

## 2. Preliminary

### 2.1. Physics-Informed Neural Networks

A typical PDE problem consists of two components: the governing equation and boundary conditions. A general PDE defined over a domain $\Omega$ with boundary $\partial\Omega$ is:

$$\mathcal{N}[u(\mathbf{x})] = \mathbf{0}, \text{for } \mathbf{x} \in \Omega, \text{ and } \mathcal{B}[u(\mathbf{x})] = \mathbf{0}, \text{ for } \mathbf{x} \in \partial\Omega,$$

where $\mathcal{N}$ is a series of differential operators, $\mathbf{x}$ represents spatial coordinates, $u$ is an unknown function and $\mathcal{B}$ is a series of constraints at the domain boundaries. PINNs use a neural network $u(\mathbf{x}; \boldsymbol{\theta})^2$ to directly approximate the solution $u^*(\mathbf{x})$ with trainable parameters $\boldsymbol{\theta}$. The training process involves minimizing a loss function that incorporates the following two components:

1. **PDE Residual Loss ($\ell_{\mathbf{r}}$)**: This term quantifies the extent to which the neural approximation violates the governing equation at interior collocation points. It is computed as the mean squared error (MSE) of the PDE residual over a set of collocation points over all derivative constraints in the domain: $\ell_{\mathrm{r}}(\boldsymbol{\theta}) = \frac{1}{N_{\mathrm{r}}} \sum_{i=1}^{N_{\mathrm{r}}} \|\mathcal{N}[u(\mathbf{x}_i; \boldsymbol{\theta})]\|_2^2$, where $\|\cdot\|_2$ is the $L_2$ norm, $\mathcal{N}[u(\mathbf{x}_i; \boldsymbol{\theta})]$ is a differential operator, generated via automatic differentiation, $\mathbf{x}_i$ are collocation points sampled from $\Omega$, and $N_{\mathrm{r}}$ is the number of residual collocation points used during PINN training.

2. **Boundary Condition Loss ($\ell_{\mathbf{BC}}$)**: This enforces the boundary conditions. It is computed as the MSE of the discrepancy between the network prediction and the boundary condition: $\ell_{\mathrm{BC}}(\boldsymbol{\theta}) = \frac{1}{N_{\mathrm{BC}}} \sum_{i=1}^{N_{\mathrm{BC}}} \|\mathcal{B}[u(\mathbf{x}_i; \boldsymbol{\theta})]\|_2^2$, where $\mathbf{x}_i$ are points sampled from the boundary $\partial\Omega$, and $N_{\mathrm{BC}}$ is the number of boundary points.

---

[1]Code is available at github.com/windindicator/CoupledNet.

[2]$\mathbf{x}$ encompasses both spatial and temporal dimensions.

The total loss function used to train the PINN is a weighted sum of these PDE-related losses: $\ell(\boldsymbol{\theta}) = \ell_{\mathrm{r}}(\boldsymbol{\theta}) + \ell_{\mathrm{BC}}(\boldsymbol{\theta})$. By minimizing this loss function, the neural network learns to approximate the solution of the PDE while satisfying the initial and boundary conditions.

## 2.2. Deep Networks Differentiation Structure Analysis

The optimization objective of PINNs fundamentally diverges from that of conventional neural networks. In PINNs, the loss function incorporates the physics of the problem, often expressed through PDE residuals. For a first-order PDE, the residual includes terms like the first-order derivative of the network output $u(\mathbf{x}; \boldsymbol{\theta})$ with respect to the input $\mathbf{x}$: $\frac{\partial u}{\partial \mathbf{x}}$. Consequently, training a PINN requires computing higher-order gradients, specifically the gradient of such derivatives with respect to the model parameters $\theta$, i.e., $\frac{\partial}{\partial \boldsymbol{\theta}}\left(\frac{\partial u}{\partial \mathbf{x}}\right)$. This contrasts sharply with conventional training, where the core computation is the first-order gradient $\frac{\partial u}{\partial \boldsymbol{\theta}}$. This necessity for gradients of derivatives highlights the critical need to analyze the network's differentiation structure within the PINN framework, particularly in deep architectures.

To understand the differentiation structure of within the PINN framework, we begin our analysis with the first-order derivatives of an MLP. Consider an MLP with $L$ hidden layers, the network output $u(\mathbf{x}; \boldsymbol{\theta})$ for input $\mathbf{x}$ is: $\mathbf{h}_i = f(\mathbf{h}_{i-1})$, for $i \geq 1$, and $\mathbf{h}_0 = \mathbf{x}$, where the $i$-th layer(or block) transforms its input $\mathbf{h}_{i-1}$ to $\mathbf{h}_i$ and the final output is $u(\mathbf{x}; \boldsymbol{\theta}) = \mathbf{h}_L$. The Jacobian matrix of the network output $u(\mathbf{x}; \boldsymbol{\theta})$ with respect to the input $\mathbf{x}$ is structured as a product of layer-wise Jacobians:

$$\frac{\partial u(\mathbf{x}; \boldsymbol{\theta})}{\partial \mathbf{x}} = \prod_{l=1}^{L} \frac{\partial \mathbf{h}_l}{\partial \mathbf{h}_{l-1}} = \prod_{l=1}^{L} \mathbf{J}_l. \tag{1}$$

This formula implies that if the norms of Jacobians are systematically smaller or larger than one, the resulting derivatives vanish or grow exponentially with depth, which causes the gradients, i.e., $\frac{\partial}{\partial \boldsymbol{\theta}}\left(\frac{\partial u}{\partial \mathbf{x}}\right)$, to vanish or explode. In Section 4, we experimentally confirm that this pathology manifests at initialization: MLPs exhibit vanishing derivatives with increasing depth, while ResNets show exploding derivative magnitudes. Wang et al. (2024a) further established a probabilistic bound revealing the intrinsic derivative vanishing pathology in MLPs with $\tanh$ activations.

The analysis in Appendix A further demonstrates that the explosion/vanishing of derivatives is distinct from conventional gradient issues, yet it directly triggers these gradient pathologies in PINN training. This highlights the need for network architectures explicitly designed to ensure stable and well-conditioned differentiation.

## 3. CoupledNet

In Section 2.2, we begin by analyzing the derivative structures of neural networks, as these derivatives are central to formulating the physical laws governed by PDEs. A key challenge in PINNs is the numerical instability during automatic differentiation, which can be traced to the properties of these derivatives. Existing studies provide crucial insights into this issue: first, the derivative of an MLP is mathematically equivalent to the successive multiplication of its weight matrices (Wang et al., 2024a); second, the spectral norm of these weight matrices governs the scaling effect on feature vectors and the magnitude of changes during gradient updates (Yang et al., 2024). Combining these insights, we deduce that the stability of the entire automatic differentiation process in PINNs is directly influenced by the spectral norms of the hidden layer Jacobian matrices. Therefore, to fundamentally stabilize this process, we propose imposing dual-bound constraints on these spectral norms. This leads to the introduction of CoupledNet, a structure designed to separately constrain the upper and lower bounds of the spectral norms, thereby stabilizing numerical behavior.

### 3.1. Coupled Block

Directly controlling the spectral norm of Jacobian matrices proves highly challenging. Without loss of generality, we assume the widths of different hidden layers are the same, denoted by $d$, for simplicity. Thus, the Jacobian matrix $\mathbf{J}_l \in \mathbb{R}^{d \times d}$ of each hidden layer is square, except for the input layer and the output layer. Under this assumption, the composite Jacobian determinant across the $L$ hidden layers (without considering the input and output layer) satisfies: $\det\left(\prod_{l=1}^{L} \mathbf{J}_l\right) = \prod_{l=1}^{L} \det(\mathbf{J}_l)$. The determinant of each Jacobian is precisely the product of its eigenvalues. We utilize the determinant of each Jacobian as an indicator. Mathematically, it can be expressed as

$$\det(\mathbf{J}_l) = \prod_{i=1}^{d} \lambda_{l,i}, \tag{2}$$

where $\lambda_{l,i}$ represents i-th eigenvalue of $\mathbf{J}_l$. As the lower bound of the spectral norm of the Jacobian matrix is the largest eigenvalue's absolute value, and thus related to the determinant of the Jacobian matrix by (2). This means that one can prevent derivatives from vanishing when $L$ is large by controlling $\det(\mathbf{J}_l) \geq 1$ for each layer $l$. Therefore, inspired by normalizing flow architectures (Dinh et al., 2017; Kingma & Dhariwal, 2018), we propose CoupledBlock as the core component of CoupledNet, which regulates the spectral norm lower bound of hidden-layer Jacobian matrices by controlling their determinants. Figure 1 presents the detailed model architecture of CoupledNet.

To enhance nonlinear transformations without disrupting

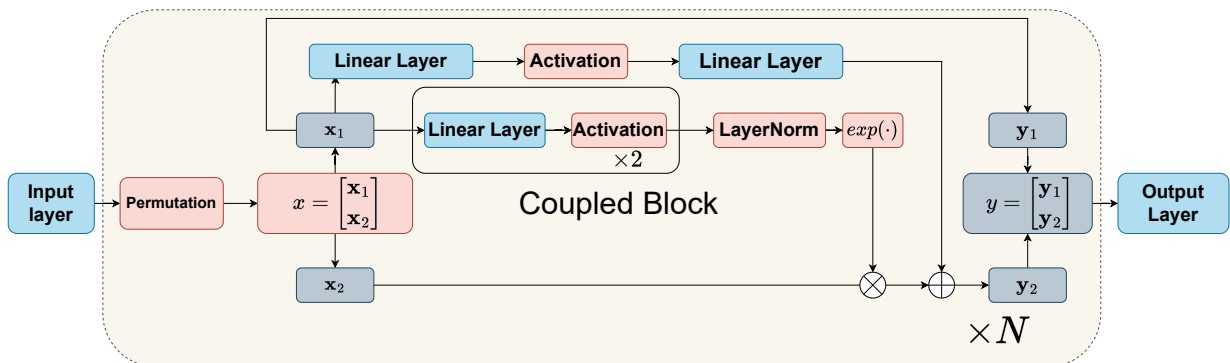

*Figure 1.* Architecture of CoupledNet: The network consists of (a) an input layer (either linear or Fourier-embedded), which elevates the original coordinates to a high-dimensional embedding space. (b) $N$ stacked Coupled Blocks for highly complex transformation. The notation "$\times 2$" denotes that two sets of linear and activation layers are stacked sequentially. (c) A linear layer maps the final output into the required output dimension.

the product of determinants, each coupled block first applies a fixed random permutation to the input, then splits it into two same-length partitions $\mathbf{x}_1$ and $\mathbf{x}_2$. The block output $[\mathbf{y}_1; \mathbf{y}_2]$ is computed via:

$$\mathbf{y}_1 = \mathbf{x}_1, \quad \mathbf{y}_2 = \mathbf{s} \odot \mathbf{x}_2 + \mathbf{t}, \tag{3}$$

where $\mathbf{s} = \exp\left(\text{LayerNorm}(\sigma(\mathbf{W}_2\sigma(\mathbf{W}_1\mathbf{x}_1 + \mathbf{b}_1) + \mathbf{b}_2))\right)$ and $\mathbf{t} = \mathbf{W}_4\sigma(\mathbf{W}_3\mathbf{x}_1 + \mathbf{b}_3) + \mathbf{b}_4$. The LayerNorm has trainable scale parameters and a frozen zero vector as the bias to make the mean of the output $\mathbf{0}$. The number of parameters of a Coupled Block is nearly identical to that of a linear layer with equal width. This architecture prevents derivative vanishing by controlling the determinant of the Jacobian matrix, with theoretical support from the following proposition.

**Proposition 3.1** (Spectral Control in CoupledNet Architecture). *The CoupledNet architecture with coupled blocks guarantees controlled spectral properties:*

1. *Each CoupledBlock's Jacobian matrix $\mathbf{J}_l$ satisfies $\|\mathbf{J}_l\|_2 \geq 1$.*

2. *Stacked CoupledBlocks maintain explicit spectral norm lower bound preservation.*

*Proof.* Proof can be found in Appendix B. □

By strictly enforcing the absolute value of the Jacobian determinant of each coupled block to be 1, we stabilize the eigenvalues of the Jacobian matrices across deep architectures. Thus, CoupledNet avoids derivative vanishing by controlling the spectral norm of the Jacobian matrix.

Moreover, the following theorem proves that CoupledNet has the potential to represent any function, indicating that

CoupledNet architecture does not limit the model's expressiveness theoretically.

**Proposition 3.2** (Universal Approximation Theorem for CoupledNet). *Let the activation function $\sigma : \mathbb{R} \to \mathbb{R}$ be a non-polynomial, bounded, and continuous function (e.g., Sigmoid, Tanh) or a piecewise linear function (e.g., ReLU). For any continuous function $f : K \to \mathbb{R}^m$ defined on a compact set $K \subset \mathbb{R}^n$, and any $\epsilon > 0$, there exists a single-Coupled-Block CoupledNet $F(x)$ such that:*

$$\sup_{x \in K} \|f(x) - F(x)\| < \epsilon,$$

*where the number of hidden neurons $N$ is sufficiently large and depends on $f$, $\epsilon$, and $K$.*

*Proof.* Proof can be found in Appendix C. □

### 3.2. Controller for The Maximum of Jacobian Spectral

We simultaneously seek a simple, effective mechanism to control the maximum spectral norm. However, the presence of permutation layers and the unbounded nature of LayerNorm operations fundamentally prevent rigorous control over the maximum spectral norm of hidden-layer Jacobian matrices. To address this limitation, we develop a two-pronged spectral containment strategy through numerical analysis of differential propagation structures in permutation-free hidden pathways. Specifically, within the CoupledBlock framework, we implement:

1. LayerNorm Spectral Containment: Bounding of directional scaling factors during feature normalization,

2. Output Layer Constraint: Direct regulation of the output of the last Coupled Block.

For any square matrices $A$ and $B$, the spectral norm of $A \cdot B$ satisfies $\|AB\| \leq \|A\| \cdot \|B\|$. Since the spectral norm of a permutation matrix is 1, multiplying by a permutation matrix does not affect the maximum control of the spectral norm. Assuming the absence of permutation layers in the worst-case path, the diagonal blocks in the Jacobian matrix product of CoupledBlock take the form:

Upper-left block: $[\mathbf{J}]_{1:\frac{n}{2}, 1:\frac{n}{2}} = \prod_{k=1}^{L} \mathbf{I}_{\frac{n}{2}} = \mathbf{I}_{\frac{n}{2}}$.

Lower-right block: $[\mathbf{J}]_{\frac{n}{2}+1:n, \frac{n}{2}+1:n} = \prod_{k=1}^{L} \operatorname{diag}(\mathbf{s}^{(k)})$, where $\left( \prod_{k=1}^{L} \operatorname{diag}(\mathbf{s}^{(k)}) \right)_{ii} = \prod_{k=1}^{L} s_i^{(k)}$, $i > \frac{n}{2}$.

Given that $\mathbf{s}$ is derived through the $\exp(\cdot)$ transformation following LayerNorm, and assuming statistical independence among elements $s_i$ (from random initialization), we impose the probabilistic constraint

$$\mathbb{P}\left( \prod_{k=1}^{L} s_i^{(k)} \leq 2 \right) \geq 0.99. \; \forall i \tag{4}$$

Through log-normal distribution analysis, the required LayerNorm standard deviation $\sigma$ satisfies $\sigma = \frac{\log 2}{z_{0.99} \cdot \sqrt{L}}$, where $z_{0.99} \approx 2.326$ denotes the 99% quantile of the standard normal distribution. We implement this derived $\sigma$ as the fixed standard deviation parameter in LayerNorm operations, thereby establishing provable upper bounds on the spectral norms of Jacobian matrices within CoupledBlock.

Simultaneously, we analyze the lower-left block of Jacobian matrices. As in the CoupledBlock controller, the exponential mapping in the transformation induces dominance of the maximum element in the lower-left Jacobian block by $max(s_i)$. When considering Jacobian matrix products across cascaded blocks, the controlled standard deviation in LayerNorm operations ensures the maximum magnitude in the composite lower-left block is characterized as a linear combination of layer-wise $max(s_i)$ values. Consequently, we implement a runtime spectral containment protocol:

1. Track $max(\mathbf{s_i})$ during forward propagation for $i$-th Coupled Block

2. Apply final normalization before output layer: $\mathbf{x} \leftarrow \frac{\mathbf{x}}{max_i(max(s_i)) * L}$.

The dynamic nature of the scaling parameter during training enables real-time numerical regulation of differentiation operators, unlike conventional initialization-based controls that remain static post-initialization. This adaptive mechanism allows continuous stabilization of differentiation structures throughout the training process. This provides CoupledNet's secondary spectral norm control mechanism. However, as this transformation directly modifies pre-activation magnitudes preceding the output layer, it inherently constrains the codomain of solution spaces. We therefore restrict its application to value-range-limited PDE classes.

## 4. PINN Depth-dependent Dynamics Analysis

To validate that CoupledNet can effectively control the norm of model derivatives by regulating spectral norms, as well as to verify that this architecture can enhance stability and strengthen performance in PINN problems through model depth extension, we conducted two experiments.

We first compared the first-order derivative norms of different base model architectures during initialization. Three models were evaluated: MLP, MLP with residual connection (ResNet-based MLP), a widely adopted solution for depth scaling in conventional deep learning (He et al., 2016), and CoupledNet. In the subsequent content, we use ResNet to denote MLP with skip connections, which is described by $\mathbf{h}_i = \sigma(\mathbf{W}\mathbf{h}_{i-1} + \mathbf{b}) + \mathbf{h}_{i-1}$. As shown in Figure 2, the MLP model exhibited an exponential decay of first-order derivatives with increasing depth due to the multiplicative structure of Jacobian matrices from consecutive linear and activation layers. While ResNet addressed vanishing derivatives, it developed derivative explosion issues as depth increased. The numerical instability caused by deeper ResNets also led to training failures in PINNs. In contrast, the CoupledNet architecture demonstrated stable first-order derivative magnitudes across varying depths, regardless of output layer scaling implementation, indicating effective resolution of numerical challenges arising from Jacobian matrix multiplication. Furthermore, since the upper-bound controller proposed in Section 3.2 was designed based on numerical analysis without considering the permutation matrix, we additionally conducted an ablation study by introducing the permutation operation in this experiment. The results show that the permutation operation had a negligible influence on the derivative norms, supporting our assumption in Section 3.2 and confirming the rationality of the Jacobian norm upper-bound controller design.

Our experiments have demonstrated that CoupledNet ensures numerical stability in derivative computations during depth escalation by enforcing both upper and lower bounds on the spectral norms of Jacobian matrices. The following experiment provides empirical evidence that CoupledNet achieves superior performance through architectural deepening. Consider an ODE defined as:

$$\frac{du(x)}{dx} = f(x); u(-2) = u_0; u(2) = u_1, \tag{5}$$

where $u(x)$ is the solution to be learned, $x \in [-2, 2]$ and $f(x)$ is the first derivative of $u(x)$. The analytical solution for $u(x)$ is $u_{\text{analytical}}(x) = \exp(x) + \frac{1}{1+x^2} + \sin(2\pi \cdot 10x)$, which is composed of the sum of three simple functions. It is easy to find that $u_{\text{analytical}}(x)$ contains the high-frequency

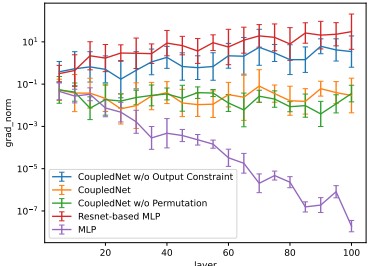
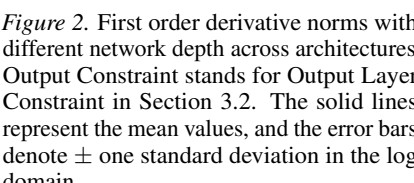

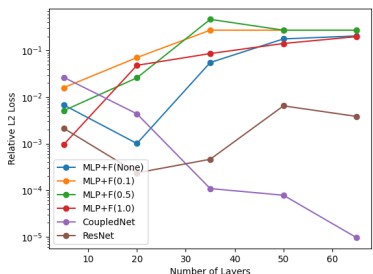

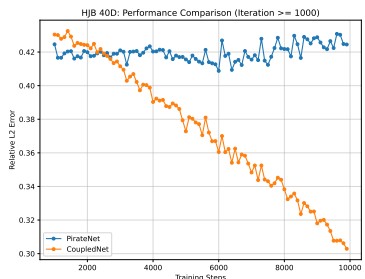

*Figure 2.* First order derivative norms with different network depth across architectures. Output Constraint stands for Output Layer Constraint in Section 3.2. The solid lines represent the mean values, and the error bars denote $\pm$ one standard deviation in the log domain.

*Figure 3.* The Figure shows the relative $L_2$ error of models with different architectures with varying depths on the target function (5). $F(m)$ stands for Fourier feature with scale $m$, and $F(\text{None})$ means we don't use Fourier feature.

*Figure 4.* **40D HJB: CoupledNet and PirateNet training curve.** Relative $L_2$ error versus training steps.

component $\sin(20\pi x)$, which is generally considered the main failure mode of PINN. In the experiment, the networks are configured with a width of 64 and trained using a batch size of 1,024 for 10,000 steps. The Adam optimizer is used with an initial learning rate of $1 \times 10^{-3}$ and the learning rate decays following an exponential schedule. We trained models with 5 random seeds for each node in Figure 3 and calculated the average of the relative $L_2$ losses.

In this experimental configuration, we benchmarked the MLP, ResNet-incorporated MLP, and CoupledNet architectures. To systematically investigate the pathological behaviors induced by network depth escalation in Multilayer Perceptron (MLP) architectures, we implement Fourier feature-enhanced MLP variants to evaluate the hypothesis that spectral embedding could mitigate depth-related pathologies. Figure 3 reveals a consistent pattern: all MLP-derived architectures exhibit progressive deterioration in training capability with increasing depth. This empirical evidence demonstrates that frequency-space transformations fail to compensate for fundamental architectural limitations in deep MLP configurations. Moreover, ResNet implementations achieve a decreasing relative $L_2$ error with depth of escalation up to 20 layers. However, this improvement trajectory reverses beyond 20 layers, with error metrics exhibiting a positive correlation with additional depth increments. These findings align with the theoretical framework proposed in (Wang et al., 2024a), confirming that skip connections alone cannot resolve the intrinsic pathologies of deep PINN architectures.

As shown in Figure 3, CoupledNet fully leverages the superior capacity of deep models in PINN training: the relative $L_2$ error decreases consistently as the model depth increases. At a depth of 50 layers, CoupledNet achieves superior prediction accuracy compared to several baseline models across all tested depths on the ODE benchmark. We conclude that

deepening a PINN with a well-designed architecture is an effective way to enhance performance when training on high-frequency information.

## 5. Experimental Results

### 5.1. High-frequency Experiments

We design this experiment to demonstrate our method's superior capability in handling complex high-frequency PDEs compared to existing approaches. The benchmark problem is defined on a 2D domain $\Omega = [-1, 1] \times [-1, 1]$ with exact solution: $u_{\text{analytical}}(x, t) = \sin\left(30 \cdot 2\pi x^2 + 15\pi t^2\right)$, where the quadratic modulation of spatial coordinates creates high-frequency patterns. The governing first-order vector PDE and boundary conditions are jointly formulated as:

$$\begin{cases} \nabla u = \mathbf{f}(x, t), & (x, t) \in \Omega \\ u(x, t) = \sin\left(30 \cdot 2\pi x^2 + 15\pi t^2\right), & (x, t) \in \partial\Omega \end{cases} \quad (6)$$

where $\nabla u = (\partial u/\partial x, \partial u/\partial t)^\top$ and $\mathbf{f}(x, t)$ is analytically derived from the exact solution.

All models are trained using the Adam optimizer (Kingma & Ba, 2015) with an initial learning rate of $1 \times 10^{-3}$ and exponential decay (period 2,000 steps, ratio 0.9) for 50,000 iterations on NVIDIA RTX 4090 GPUs with the JAX (Bradbury et al., 2018) framework. And the batchsize is 1024. We compare two state-of-the-art baselines: the modified MLP in JaxPi (Wang et al., 2023) and PirateNet (Wang et al., 2024a). Both models utilize Fourier features (Wang et al., 2021b), random matrix factorization (Wang et al., 2022b), causal training (Wang et al., 2024b) and NTK reweighting (Wang et al., 2022c) techniques. Hyperparameter grids are searched for all methods (details are in Appendix F).

The experimental results are shown in Table 1. CoupledNet demonstrates superior performance in solving high-

*Table 1.* Metrics of Three models (Jaxpi, PirateNet, and Coupled-Net) on PDEs (6) (Best results are highlighted in bold).

| Model | BL | RL | Relative $L_2$ |
|---|---|---|---|
| Pirate | **1.8e-4** | 4.98 | 0.0334 |
| Jaxpi | 1.0e-3 | 4.31 | 0.0324 |
| Pirate w/o NTK | 1.4e-2 | 1689.5 | 0.0785 |
| Jaxpi w/o NTK | 2.4e-2 | 40.38 | 0.1022 |
| CoupledNet (8-layer) | 2.9e-4 | **0.9432** | **0.0117** |

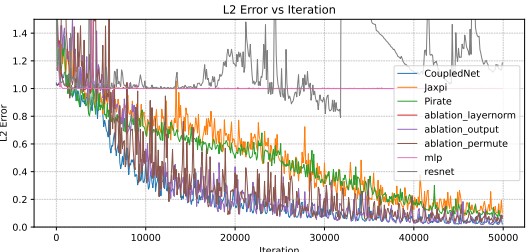

*Figure 5.* Ablation study on (6). We track the evolution of the relative $L_2$ error across training steps. The study investigates the effects of the two upper-bound controllers and the permutation matrix. CoupledNet exhibits the fastest convergence. "ablation" refers to the removal of the specified component from CoupledNet. Specifically, "layernorm" refers to removing the LayerNorm Spectral Containment from Section 3.2, "output" refers to removing the Output Layer Constraint from Section 3.2, and "-permute" refers to removing the permutation layers.

*Table 3.* Relative $L_2$ error on Eq. (7) under shallow vs. deep configurations.

| Model | Shallow | Deep |
|---|---|---|
| CoupledNet | 0.5775 (8-layer) | **0.0718** (16-layer) |
| PirateNet | 2.673 (9-layer) | 4.067 (18-layer) |

frequency PDEs. As shown in Table 1, both the final relative $L_2$ loss and residual loss of CoupledNet achieve the lowest values among the three compared models. While its boundary loss is slightly higher than PirateNet's, we attribute this phenomenon to the NTK reweighting technique employed by both PirateNet and Jaxpi during training, which helps maintain appropriate loss weights even with reduced boundary loss magnitudes. Compared with PirateNet and Jaxpi models that do not use NTK re-weighting, CoupledNet has far lower Boundary Loss and Residual Loss than the two. During the hyperparameter search process, we did not perform any reweighting for CoupledNet. However, CoupledNet still achieved a good balance between boundary loss and residual loss. We hypothesize that this advantage stems from CoupledNet's unique differentiation architecture, which intrinsically enhances gradient learning efficiency. The visualization results are shown in Appendix E.

Table 2 reports a depth ablation of CoupledNet on the same high-frequency PDE, with the corresponding hyperparameters listed in Appendix Table 7. The relative $L_2$ error remains low from 8 to 24 layers, showing that Coupled-Net can be deepened without optimization collapse. This ablation supports the depth stability of CoupledNet.

*Table 2.* Depth ablation of CoupledNet on PDE (6).

| Depth | 4 | 8 | 16 | 24 |
|---|---|---|---|---|
| Relative $L_2$ | 0.0200 | 0.0117 | 0.0147 | **0.0116** |

To further examine the convergence behavior of Coupled-Net and assess the effects of its two upper-bound controllers and the permutation matrix, we performed an ablation study using the same PDE setup. We tracked the evolution of the relative $L_2$ error, boundary loss, and residual loss over training steps for four models: MLP, ResNet, PirateNet, Jaxpi, and CoupledNet. As shown in Figure 5 and Figures in Appendix I, MLP and ResNet fail to converge, which aligns with the pathological behavior predicted by our analysis. CoupledNet exhibits the fastest convergence among all models and naturally maintains a balance between residual and boundary losses, without any reweighting strategy. Notably, when the permutation matrix is removed from CoupledNet, the final relative $L_2$ error increases to 0.063, confirming the effectiveness of the permutation matrix design.

In summary, CoupledNet delivers superior performance on high-frequency PDEs, revealing its inherent potential for addressing the learning challenges posed by high-frequency PDE systems through architectural innovations.

### 5.2. High-dynamic-range Experiments

As both PirateNet and CoupledNet are depth-enhancing frameworks for PINNs, we conduct a comparative analysis of their architectural efficacy under equivalent depth configurations. To assess their capacity to approximate complex analytic solutions, we still employ a first-order PDE system:

$$\begin{cases} \nabla u = \mathbf{f}(x,t), & (x,t) \in \Omega \\ u(x,t) = e^{5\sin\left(2\pi\sin\left(\pi(x^2+2t^2)\right)\right)}, & (x,t) \in \partial\Omega \end{cases} \quad (7)$$

where $\Omega = [-1,1] \times [-1,1]$, and $u_{\text{analytical}}(x,t) = e^{5\sin\left(2\pi\sin\left(\pi(x^2+2t^2)\right)\right)}$.

*Table 4.* Relative $L_2$ error on the HJB problem under different input dimensions (Best results are highlighted in bold).

| Method | Dimension | | | |
|---|---|---|---|---|
| | 25 | 30 | 35 | 40 |
| **CoupledNet** | **0.082** | **0.106** | **0.075** | **0.137** |
| **PirateNet** | 0.427 | 0.393 | 0.426 | 0.400 |
| **MLP** | 0.128 | 0.198 | 0.217 | 0.240 |

In this experiment, both CoupledNet and PirateNet learn with the same hyperparameters, which are presented in Ap-



*Figure 6.* Visualization of CoupledNet predictions, analytical solutions, and absolute errors for the Advection equation.

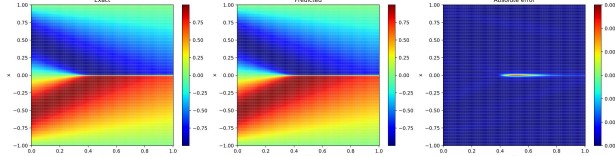

*Figure 8.* Visualization of CoupledNet predictions, analytical solutions, and absolute errors for the Burgers equation.

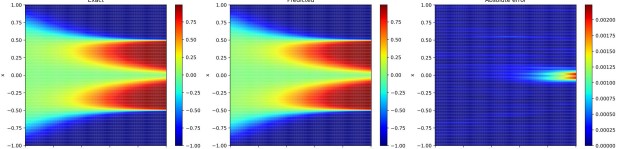

*Figure 7.* Visualization of CoupledNet predictions, analytical solutions, and absolute errors for the Allen–Cahn equation.



*Figure 9.* Visualization of CoupledNet predictions, analytical solutions, and absolute errors for the Grey-Scott equation.

pendix G. Table 3 shows that CoupledNet benefits substantially more from increased depth on this challenging PDE, achieving much lower error than PirateNet in both shallow and deep settings; although CoupledNet incurs higher wall-clock cost per step, the improved accuracy and convergence make this overhead a reasonable trade-off for this task.

The models' visualization results are shown in Appendix D. Both PirateNet and CoupledNet can obtain more accurate results merely by deepening the model. This phenomenon supplements the conclusion we reached in Section 4: *Deepening the model not only leads to more accurate results in ODE problems but also holds true for PDEs.*

Existing studies have identified spectral bias in PINN training, where models preferentially learn low-frequency components over high-frequency features (Rahaman et al., 2019). However, our experimental findings provide new insights into this phenomenon. As shown in Appendix D, the analytic solution's high-frequency features concentrate within concentric annuli, separated by low-frequency regions. Both CoupledNet and PirateNet predictions exhibit partial neglect of low-frequency components during learning.

This observation leads us to hypothesize that attributing PINN training pathologies solely to high-frequency characteristics constitutes an oversimplification. The underlying challenges may involve additional factors beyond frequency domain considerations, including:1) Magnitude disparities across different spatial regions of the analytic solution; 2) Numerical characteristics of solution derivatives; 3) Relative scale relationships between the solution and its differential terms. Our results suggest that frequency-based analysis should be augmented with complementary perspectives to fully understand PINN training dynamics.

### 5.3. High-dimensional Experiments

To more thoroughly stress-test the architectural properties of CoupledNet, we conduct experiments on a challenging high-dimensional PDE. Specifically, we consider the Hamilton–Jacobi–Bellman (HJB) equation from (Wang et al., 2022a) and evaluate CoupledNet across multiple input dimensions to examine its behavior in high-dimensional regimes. The full PDE specification and the detailed hyperparameter configurations are provided in Appendix H.1 and Appendix F (Table 8). We compare CoupledNet with PirateNet, MLP and report relative $L_2$ error in Table 4. As shown in Table 4, CoupledNet achieves the lowest relative $L_2$ error in the higher-dimensional HJB settings.

We additionally analyze the training dynamics in the high-dimensional HJB setting. At dimension $d = 40$, Figure 4 (on Page 6) plots the relative $L_2$ error versus training steps for CoupledNet and PirateNet. The curves show that CoupledNet yields a steady reduction of the error throughout training, whereas PirateNet do not exhibit a similarly consistent decreasing trend and instead remains unstable in this configuration. This behavior is consistent with our design motivation: CoupledNet introduces a more structured differentiation pathway, which may help maintain more stable derivative propagation and improve optimization stability on challenging high-dimensional PDEs.

*Table 5.* Results on Jaxpi Benchmark (Best results are highlighted in bold).

| PDE | Jaxpi | PirateNet | CoupledNet |
|---|---|---|---|
| Advection | 6.88e-4 | 4.88e-4 | **3.64e-4** |
| Allen-Cahn | 5.37e-5 | **2.24e-5** | 9.09e-5 |
| Burgers | 1.43e-4 | **4.03e-5** | 1.22e-4 |
| Grey-Scott | 6.13 | **3.61e-3** | 1.93e-2 |
| Korteweg-De Vries | 1.96e-3 | **4.27e-4** | 1.30e-3 |

## 5.4. Results on Conventional PDEs

To validate CoupledNet's performance on conventional PDE benchmarks, we conduct experiments using the benchmark suite from Jaxpi (Wang et al., 2023), with full PDE specifications and hyperparameter settings documented in Appendix H. As shown in Table 5, CoupledNet gets competitive results on the benchmarks. Importantly, even on relatively simple PDEs where CoupledNet is not the best in relative metrics, it still captures the key solution structures with very small absolute pointwise errors (see Figures 6, 7, 8 and 9). To further analyze the training dynamics, we examine the evolution of the neural tangent kernel (NTK) spectrum during training on the 1D Poisson equation from Wang et al. (2021b). In Appendix K, we further present heatmaps demonstrating that, at early stages of training, the eigenvector corresponding to the top-3 NTK eigenvalue in CoupledNet exhibits pronounced high-frequency components, whereas MLP and PirateNet tend to capture such high-frequency signatures only after substantially longer optimization. Improving the computational efficiency of CoupledNet is an important direction for future research.

## 6. Conclusion

Current deep learning architectures, such as MLPs and ResNets, exhibit initialization pathologies that fundamentally limit their depth scalability. To address this, we propose CoupledNet, a novel physics-informed deep learning architecture. CoupledNet achieves state-of-the-art performance on PDE systems with complex analytical solutions. Although CoupledNet does not achieve state-of-the-art results on all PDE benchmarks, it remains competitively accurate. By addressing the limitations of existing approaches and providing a stable framework for deep PINNs, our work opens new avenues for applying deep learning to complex physical systems.

## Impact Statement

This paper presents work whose goal is to advance the field of Machine Learning and Scientific Computation. There are many potential societal consequences of our work, none of which we feel must be specifically highlighted here.

## Acknowledgments

The research work described in this paper was conducted in the JC STEM Lab of Machine Learning and Symbolic Reasoning funded by The Hong Kong Jockey Club Charities Trust. We would also like to thank the reviewers for their thorough reading of our manuscript and for their constructive suggestions, which have undoubtedly improved the final version.

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

## A. Simple Example

The loss function in PINNs typically comprises two parts: a data-driven term (e.g., for boundary/initial conditions) and a physics-driven term (e.g., for the governing ODE/PDE). The gradient computation for the data-driven term mirrors that of traditional neural networks. However, the gradient of the physics-driven term is directly governed by the derivatives of the differential equation. To explicitly illustrate this critical relationship between the network's gradient and the equation's derivatives, we consider a simplified case of a static ODE:

$$\frac{\partial u}{\partial \mathbf{x}} = f(\mathbf{x})$$

The boundary loss is a typical supervised learning problem that is well-established in the deep learning literature. We consider the other part of the loss function in PINNs, using the vanilla residual loss as an example, to show where the difference and difficulty come from:

$$L = \int (\frac{\partial u(\mathbf{x})}{\partial \mathbf{x}} - f(\mathbf{x}))^2 dx \approx \frac{1}{N} \sum_{i=1}^{N} \left( \frac{\partial u(\mathbf{x}_i)}{\partial \mathbf{x}_i} - f(\mathbf{x}_i) \right)^2$$

The gradient of a single data point of the residual loss is:

$$\frac{\partial L}{\partial \theta} = 2 \left( \frac{\partial u(\mathbf{x}_i)}{\partial \mathbf{x}_i} - f(\mathbf{x}_i) \right) \frac{\partial}{\partial \theta} \left( \frac{\partial u(\mathbf{x}_i)}{\partial \mathbf{x}_i} \right)$$

Considering the gradient of parameters at the $i$-th layer:

$$\frac{\partial L}{\partial \theta_i} = 2 \left( \frac{\partial u(\mathbf{x}_j)}{\partial \mathbf{x}_j} - f(\mathbf{x}_j) \right) \frac{\partial}{\partial \theta_i} \left( \frac{\partial u(\mathbf{x}_j)}{\partial \mathbf{x}_j} \right)$$
$$= r_j \left[ \sum_{m=i}^{L} \left( \frac{\partial u}{\partial h_m} \cdot \frac{\partial^2 h_m}{\partial \theta_i \partial h_{m-1}} \cdot \frac{\partial h_{m-1}}{\partial \mathbf{x}} \right) \right],$$

where $r_j = \frac{\partial u(\mathbf{x}_j)}{\partial \mathbf{x}_j} - f(\mathbf{x}_j)$ represents the residual error.

The gradient depends on both the forward propagation path ($\frac{\partial h_{m-1}}{\partial \mathbf{x}}$) and the backward propagation path ($\frac{\partial u}{\partial h_m}$), connected through the second-order derivative $\frac{\partial^2 h_m}{\partial \theta_i \partial h_{m-1}}$. $\frac{\partial u}{\partial h_m}$ is the product of Jacobian matrices from layer $m$ to the output layer $L$ $\frac{\partial h_{m-1}}{\partial \mathbf{x}}$ is the product of Jacobian matrices from the input layer to layer $m-1$.

For parameters in intermediate layers ($1 < i < L$), the gradient computation involves Jacobian products that span nearly the entire network architecture, in contrast to traditional backpropagation, where parameters only face Jacobians from their layer to the output. This analytical result demonstrates that PINN gradients inherently encode richer architectural information than conventional neural networks, as they must account for both the forward flow of input variations and the backward flow of output sensitivities simultaneously.

Therefore, PINN's gradient problem has fundamental differences from traditional, well-known gradient problems. For sufficiently deep models, even for parameters close to the output layer, numerical issues in the Jacobian matrices of layers before them will still affect the updates of these parameters.

Therefore, we can conclude that the numerical issues of model derivatives are an important cause of PINN gradient problems. PINN's derivative numerical problems and gradient numerical problems are not independent; when model derivatives are numerically unstable, PINN optimization will not obtain stable gradients.

## B. Proof of Proposition 3.1

*Proof.* We establish these properties through an analytic examination of the network's differential structure:

*Part 1: Spectral norm of Coupled Block.*

Consider the coupled block's Jacobian decomposition:

$$\mathbf{J}_l = \mathbf{J'}_l \cdot \mathbf{P}, \text{ with } \mathbf{J'}_l = \begin{bmatrix} \mathbf{I}_l & \mathbf{0} \\ \text{diag}(\mathbf{x}_2)\mathbf{J}_s + \mathbf{J}_t & \text{diag}(\mathbf{s}) \end{bmatrix}. \tag{8}$$

The triangular structure of $\mathbf{J'}_l$ yields determinant:

$$\det(\mathbf{J'}_l) = \prod_{i=1}^{d} s_i = \exp(\sum \log s_i) = \exp(0) = 1, \tag{9}$$

due to the LayerNorm constraint with no bias added, see Section 3.

Since $P$ is a permutation matrix, each layer maintains:

$$|\det(\mathbf{J}_l)| = |\det(\mathbf{J'}_l)||\det(\mathbf{P})| = 1 \tag{10}$$

Therefore, we have

$$\|\mathbf{J}_l\|_2 \geq \rho(\mathbf{J}_l) \geq 1. \tag{11}$$

*Part 2: Lower bound of the Spectral norm of the Jacobian of CoupledNet.*

For $L$-layer network(CoupledNet) with total Jacobian $\mathbf{J} = \prod_{l=1}^{L} \mathbf{J}_l$:

$$\det(\mathbf{J}) = \prod_{l=1}^{L} \det(\mathbf{J}_l) = 1. \tag{12}$$

From spectral theory, the spectral radius $\rho(\mathbf{J})$ satisfies:

$$\rho(\mathbf{J}) = \max_{1 \leq k \leq n} |\lambda_k| \geq |\det(\mathbf{J})|^{1/n} = 1. \tag{13}$$

Relating spectral radius to spectral norm:

$$\|\mathbf{J}\|_2 \geq \rho(\mathbf{J}) \geq 1. \tag{14}$$

This completes the proof. □

## C. Proof of Proposition 3.2

*Proof.* We demonstrate a degenerate case of CoupledNet, which possesses the universal approximation ability, thereby showing that CoupledNet can approximate any continuous function. Let the input layer be a fully-connected layer without bias:

$$\mathbf{x}' = \mathbf{W}_{\text{input}}x, \quad \mathbf{W}_{\text{input}} \in \mathbb{R}^{N \times n} \tag{15}$$

Suppose that the permutation layer of the single-Coupled-Block CoupledNet is an identity function. Then, we have

$$F(x) = \mathbf{W}_{\text{output}}\mathbf{y} + \mathbf{b}_{\text{output}}, \tag{16}$$

where y is the output of the Coupled Block:

$$\mathbf{y} = \begin{bmatrix} \mathbf{y}_1 \\ \mathbf{y}_2 \end{bmatrix}, \mathbf{y}_1 = \mathbf{x'}_1, \mathbf{y}_2 = \mathbf{s} \odot \mathbf{x'}_2 + \mathbf{t}, \tag{17}$$

where $\mathbf{s} = \exp(\text{LayerNorm}(\sigma(\mathbf{W}_2\sigma(\mathbf{W}_1\mathbf{x'}_1 + \mathbf{b}_1) + \mathbf{b}_2)))$, $\mathbf{t} = \mathbf{W}_4\sigma(\mathbf{W}_3\mathbf{x'}_1 + \mathbf{b}_3) + \mathbf{b}_4$, and $\mathbf{x'} = \begin{bmatrix} \mathbf{x'}_1 \\ \mathbf{x'}_2 \end{bmatrix}$.

Let $\mathbf{W}_2$ be a $\mathbf{0}$ matrix, then we have $\mathbf{s} = \mathbf{1}^{\frac{N}{2}}$. Moreover, let $\mathbf{W}_3$ and $\mathbf{W}_4$ be $\mathbf{I}^{\frac{N}{2}}$ and $\mathbf{b}_4$ equal to $-\mathbf{x'}_2$, we can derive:

$$F(x) = \mathbf{W}_{\text{output}} \cdot \begin{bmatrix} \mathbf{x'}_1 \\ \sigma(\mathbf{x'}_1 + \mathbf{b}_3) \end{bmatrix} + \mathbf{b}_{\text{output}}. \tag{18}$$

Let $\mathbf{W}_{\text{output}} = \left[ \mathbf{0}^{m \times \frac{N}{2}} | \mathbf{W}'_{\text{output}} \right]$, we have

$$F(x) = \mathbf{W}'_{\text{output}} \sigma (\mathbf{W}'_{\text{input}} x + \mathbf{b}_3) + \mathbf{b}_{\text{output}}, \tag{19}$$

where $\mathbf{W}'_{\text{input}}$ is the upper half of $\mathbf{W}_{\text{input}}$.

By the Universal Approximation Theorem for MLP from (Hornik, 1991), we can drive that for any continuous function $f : K \to \mathbb{R}^m$ defined on a compact set $K \subset \mathbb{R}^n$, and any $\epsilon > 0$, there exists a single-Coupled-Blosck CoupledNet $F(x)$ such that:

$$\sup_{x \in K} \|f(x) - F(x)\| < \epsilon, \tag{20}$$

where the number of hidden neurons $N$ is sufficiently large and depends on $f$, $\epsilon$, and $K$. $\qquad\square$

## D. Figure of Comparison Between PirateNet and CoupledNet

Figure 10 visualizes the predictions of CoupledNet and PirateNet on the PDE in Eq. (7) under different depth settings. As the depth increases, CoupledNet yields predictions that more closely match the analytical solution, while PirateNet remains noticeably less accurate even with additional layers. The error maps in the bottom row further show that the 16-layer CoupledNet attains substantially smaller absolute errors than the 18-layer PirateNet.

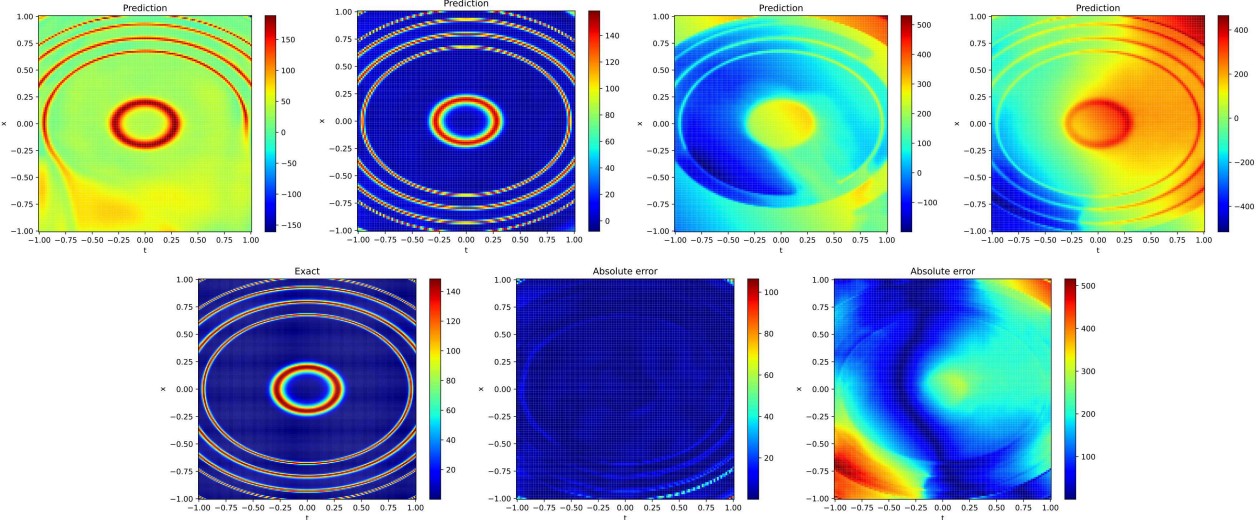

*Figure 10.* Comparison between CoupledNet and Pirate on the PDE in (7). **Top**: From left to right are the prediction results of an 8-layer CoupledNet, a 16-layer CoupledNet, a 9-layer PirateNet, and an 18-layer PirateNet. **Bottom**: From left to right are the analytical solution of the PDE, the absolute error of the predicted solution by the 16-layer CoupledNet, and the absolute error of the predicted solution by the 18-layer PirateNet.

## E. Visualization Results of Section 5.1

Figure 11 compares the analytical solution of the PDE in Eq. (6) with the predictions produced by CoupledNet, Jaxpi, and PirateNet. CoupledNet produces a solution field that is visually closer to the exact reference. This qualitative comparison complements the quantitative results in Section 5.1.

## F. Hyperparameter Configuration for Section 5.1

All experiments reported in Section 5.1 and Figure 11 share a unified hyperparameter search grid, as summarized in Table 6, while the final configurations for each method are selected from this grid. Specifically, we consider multilayer perceptron architectures with varying depths and hidden dimensions depending on the model. For PirateNet, the number of layers is chosen from $\{9, 18\}$, while Jaxpi uses $\{4, 8\}$ layers and CoupledNet adopts $\{8, 16\}$ layers. The hidden size is selected from

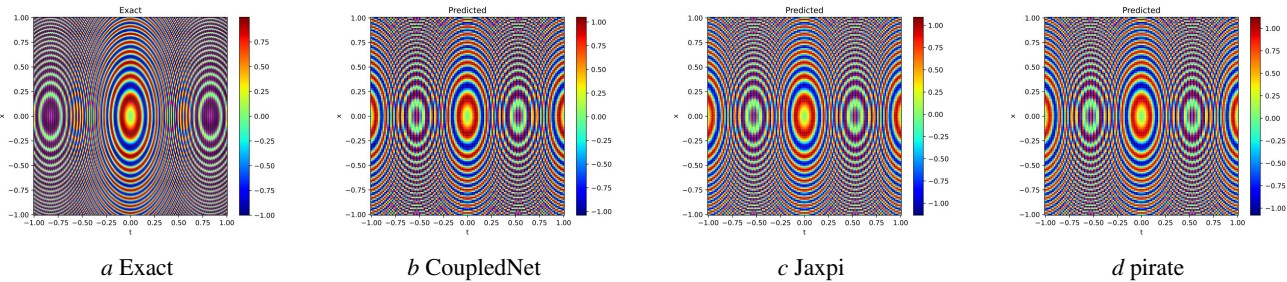

*a* Exact          *b* CoupledNet          *c* Jaxpi          *d* pirate

*Figure 11.* Comparison between the solution predicted by three models and the analytical solution on PDE system (6).

*Table 6.* Hyperparameter Search Grid for experiment in Section 5.1 three methods for results in Figure 11.

| Parameter | Value |
|---|---|
| **Architecture** | |
| Number of layers | 9, 18 for PirateNet, |
| | 4, 8 for Jaxpi |
| | 8, 16 for CoupledNet |
| Hidden size | 128, 256 |
| Activation function | tanh |
| Fourier embedding scale | NA, 1.0, 2.0, 4.0 |
| Random weight factorization | $(\mu = 0.5, \sigma = 0.1)$, |
| | $(\mu = 1.0, \sigma = 0.1)$ |
| Learning-rate decay | exponential, cosine |
| | |
| **Weighting** | |
| Weighting scheme | NA for CoupledNet |
| | NA, ntk, grad norm |
| | for Jaxpi and PirateNet |
| Warm up step | 0,2000 |
| Causal tolerance | NA, 1.0 |
| Number of Chunks(if use Causal) | 32 |

$\{128, 256\}$, and all models employ the `tanh` activation function. When applicable, Fourier feature embeddings are used with scaling factors in $\{1.0, 2.0, 4.0\}$, and random weight factorization is applied with mean $\mu \in \{0.5, 1.0\}$ and standard deviation $\sigma = 0.1$. The learning-rate decay schedule is selected from exponential decay and cosine decay.

Regarding the weighting strategy, PirateNet and Jaxpi support multiple schemes, including no weighting, neural tangent kernel (NTK) weighting, and gradient normalization, whereas CoupledNet does not employ any weighting mechanism. For the final experiments, both PirateNet and Jaxpi use NTK weighting with no warm-up steps, while CoupledNet does not apply weighting and uses a warm-up period of 2000 steps. Causal tolerance and chunk-based causal training are not activated in the reported results; however, when causal training is enabled in the search phase, the number of chunks is fixed to 32. The final selected configurations are 18 layers for PirateNet, 4 layers for Jaxpi, and 8 layers for CoupledNet, all with a hidden size of 256, Fourier embedding scale 4.0, and random weight factorization parameters $\mu = 0.5$ and $\sigma = 0.1$.

## G. Hyper-parameter in Section 5.2

Neither CoupledNet nor PirateNet undergoes hyperparameter tuning, and neither NTK re-weighting nor Fourier features are used. The hyperparameters are selected based on those employed in PirateNet experiments across multiple PDEs (Wang et al., 2024a). The optimizer chosen is Adam, with an initial learning rate of $1 \times 10^{-3}$ for 8-layer CoupledNet (9-layer PirateNet) and $1 \times 10^{-4}$ for 16-layer CoupledNet (18-layer PirateNet), using exponential decay (period 2,000 steps, ratio 0.9). Since the 18-layer PirateNet failed to converge with a learning rate of $1 \times 10^{-4}$, we conducted additional experiments using a learning rate of $1 \times 10^{-3}$. The width of the hidden layers is 256, the training step is 50000, the activation function is tanh, and the batch size is 512. Because the problem has a large value range, CoupledNet did not use the output layer constraint in this experiment.

*Table 7.* Hyperparameters for the CoupledNet depth ablation in Table 2.

| Depth | 4 | 8 | 16 | 24 |
|---|---|---|---|---|
| Hidden size | 128 | 128 | 128 | 128 |
| Activation | tanh | tanh | tanh | tanh |
| Fourier scale | 4.0 | 4.0 | 4.0 | 4.0 |
| Fourier dim | 128 | 128 | 128 | 128 |
| Weight factorization $(\mu, \sigma)$ | $(0.5, 0.1)$ | $(0.5, 0.1)$ | $(0.5, 0.1)$ | $(0.5, 0.1)$ |
| Optimizer | Adam | Adam | Adam | Adam |
| Learning rate | $10^{-3}$ | $10^{-3}$ | $10^{-3}$ | $10^{-3}$ |
| LR schedule | cosine | cosine | cosine | cosine |
| Cosine decay steps | 50k | 50k | 50k | 50k |
| Cosine $\alpha$ | 0.01 | 0.01 | 0.01 | 0.01 |
| Warm-up steps | 2000 | 0 | 2000 | 0 |
| Batch size | 1024 | 1024 | 1024 | 1024 |
| Training steps | 50k | 50k | 50k | 50k |
| Weighting | none | none | none | none |
| Causal training | no | no | no | no |

*Table 8.* Configuration Details of CoupledNet on Advection equation

| Category | Advection equation | Allen-Cahn equation | GS equation | KdV equation | Burgers equation | HJB equation |
|---|---|---|---|---|---|---|
| Architecture Name | CoupledMlp | CoupledMlp | CoupledMlp | CoupledMlp | CoupledMlp | CoupledMlp |
| Number of Layers | 9 | 9 | 8 | 32 | 3 | 3 |
| Hidden Dimension | 256 | 256 | 256 | 256 | 256 | 128 |
| Output Dimension | 1 | 1 | 2 | 1 | 1 | 1 |
| Activation Function | tanh | tanh | swish | tanh | tanh | tanh |
| Periodicity | $(\pi, 1.0)$ | $(\pi)$ | $(\pi,\pi)$ | $(\pi,)$ | None | None |
| axis: | (0, 1) | (0, ), | (1, 2) | (1,) | None | None |
| trainable: | (True, False) | (False, ) | (False, False) | (False,) | None | None |
| Fourier Embedding | embed-scale: 1.0 | 2.0 | 1.0 | 1.0 | 1.0 | None |
|  | embed-dim: 256 | 256 | 256 | 256 | 256 | None |
| Reparameterization | None | None | $(\mu = 0.5, \sigma = 0.1)$ | None | None | $(\mu = 1.0, \sigma = 0.1)$ |
| PI Initialization | None | None | True | None | None | None |
| Optimizer | Adam | Adam | Adam | Adam | Adam | Adam |
| Beta1 | 0.9 | 0.9 | 0.9 | 0.9 | 0.9 | 0.9 |
| Beta2 | 0.999 | 0.999 | 0.999 | 0.999 | 0.999 | 0.999 |
| Epsilon | 1e-8 | 1e-8 | 1e-8 | 1e-8 | 1e-8 | 1e-8 |
| Learning Rate | 1e-3 | 1e-3 | 1e-3 | 1e-3 | 1e-3 | 1e-3 |
| Decay Rate | 0.9 | 0.9 | 0.9 | 0.9 | 0.9 | 0.9 |
| Decay Steps | 2000 | 5000 | 2000 | 2000 | 1000 | 2000 |
| Staircase | False | False | False | False | False | False |
| Warmup Steps | 5000 | 5000 | 5000 | 5000 | 5000 | 0 |
| Gradient Accumulation Steps | 0 | 0 | 0 | 0 | 0 | 0 |
| Max Training Steps | 200000 | 300000 | 100000 | 200000 | 100000 | 20000 |
| Batch Size per Device | 2048 | 4096 | 4096 | 4096 | 8192 | 2048 |
| Weighting Scheme | ntk | ntk | grad_norm | grad_norm | grad_norm | none |
| Initial Weights | RL: 1.0 | 1.0 | (1.0, 1.0) | 1.0 | 1.0 | 1.0 |
|  | BL: 1.0 | 1.0 | (1.0, 1.0) | 1.0 | 1.0 | 10.0 |
| Momentum | 0.9 | 0.9 | 0.9 | 0.9 | 0.9 | 0.9 |
| Update Every Steps | 1000 | 1000 | 1000 | 1000 | 1000 | 1000 |
| Use Causal | True | True | True | True | True | False |
| Causal Tolerance | 1.0 | 1.0 | 1.0 | 0.1 | 1.0 | 0.01 |
| Time window | None | None | 10 | None | None | None |
| Number of Chunks | 32 | 32 | 32 | 16 | 32 | 32 |

# H. Hyper-Parameter in PDE benchmark

For these benchmarks, we used the experimental setups of Jaxpi and PirateNet (Wang et al., 2023; 2024a). We use the published open source data from Jaxpi as training and testing data in these benchmarks. `https://github.com/PredictiveIntelligenceLab/jaxpi`

## H.1. Hamilton-Jacobi-Bellman equation

Hamilton-Jacobi-Bellman (HJB) equation used in the experiment is:

$$\begin{cases} \mathcal{L}_{\text{HJB}} u := \partial_t u(x,t) + \frac{1}{2}\sigma^2 \Delta u(x,t) - \sum_{i=1}^n A_i |\partial_{x_i} u|^{c_i} = \varphi(x,t), & (x,t) \in \mathbb{R}^n \times [0,T] \\ \mathcal{B}_{\text{HJB}} u := u(x,T) = g(x), & x \in \mathbb{R}^n \end{cases}$$

where $A_i = \underbrace{(a_i\alpha_i)^{-\frac{1}{\alpha_i-1}}}_{\text{First term}} - \underbrace{a_i(a_i\alpha_i)^{-\frac{\alpha_i}{\alpha_i-1}}}_{\text{Second term}} \in (0,+\infty)$ and $c_i = \frac{\alpha_i}{\alpha_i-1} \in (1,+\infty)$. Hyperparameters are shown in Table 8.

## H.2. Advection equation hyper-parameters

The advection equation used in the experiment is:

$$\frac{\partial u}{\partial t} + c\frac{\partial u}{\partial x} = 0, \quad t \in [0,1], x \in (0,2\pi)$$
$$u(0,x) = g(x), \quad x \in (0,2\pi)$$

where $c = 80$ and $g(x) = \sin(x)$. Hyper-parameters are shown in Table 8

## H.3. Allen-Cahn equation

Allen-Cahn equation used in the experiment is:

$$u_t - 0.0001u_{xx} + 5u^3 - 5u = 0, \quad t \in [0,1], x \in [-1,1]$$
$$u(0,x) = x^2\cos(\pi x)$$
$$u(t,-1) = u(t,1)$$
$$u_x(t,-1) = u_x(t,1).$$

Hyperparameters are shown in Table 8.

## H.4. Grey-Scott equation

Grey-Scott equation used in the experiment is:

$$\begin{cases} u_t = \epsilon_1 \Delta u + b_1(1-u) - c_1 uv^2, & t \in (0,2), (x,y) \in (-1,1)^2 \\ v_t = \epsilon_2 \Delta v - b_2 v + c_2 uv^2, & t \in (0,2), (x,y) \in (-1,1)^2 \end{cases}$$

subject to the periodic boundary conditions and the initial conditions

$$\begin{cases} u_0(x,y) = 1 - \exp\left(-10\left((x+0.05)^2 + (y+0.02)^2\right)\right), \\ v_0(x,y) = 1 - \exp\left(-10\left((x-0.05)^2 + (y-0.02)^2\right)\right). \end{cases}$$

Hyperparameters are shown in Table 8.

## H.5. Korteweg–De Vries equation

Korteweg–De equation used in the experiment is:

$$u_t + \eta u u_x + \mu^2 u_{xxx} = 0, t \in (0,1), x \in (-1,1)$$
$$u(x,0) = \cos(\pi x)$$
$$u(t,-1) = u(t,1),$$

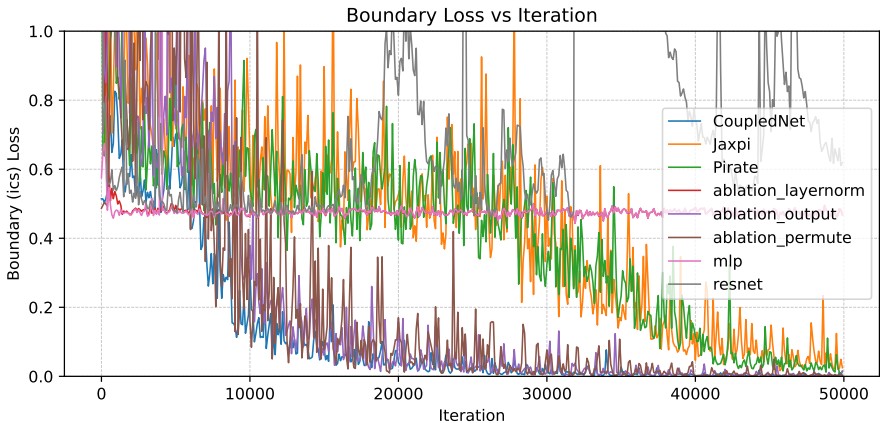

*Figure 12.* Ablation study on Equation 6.

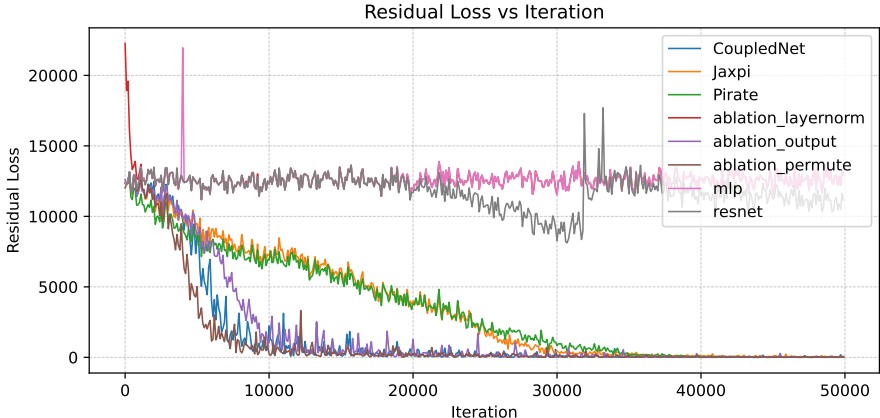

*Figure 13.* Ablation study on the Equation 6.

where $\eta = 1$ and $\mu = 0.022$ Hyperparameters are shown in Table 8.

### H.6. Burger's equation

Burger's equation used in the experiment is:

$$\frac{\partial u}{\partial t} + \frac{\partial u}{\partial x} = \frac{0.01}{\pi} \frac{\partial^2 u}{\partial x^2}$$

Hyperparameters are shown in Table 8.

## I. Ablation study results on High Frequency PDE

We track the evolution of boundary condition loss across training steps. "ablation" indicates the removal of the specified component from CoupledNet. Specifically, "layernorm" refers to removing the LayerNorm Spectral Containment from Section 3.2, "output" refers to removing the Output Layer Constraint from Section 3.2, and "-permute" refers to removing the permutation layers.

The evolution of residual loss across training steps. "ablation" indicates the removal of the specified component from CoupledNet. Specifically, "layernorm" refers to removing the LayerNorm Spectral Containment from Section 3.2, "output" refers to removing the Output Layer Constraint from Section 3.2, and "-permute" refers to removing the permutation layers.

## J. Jaxpi Benchmark Visualization Results

This section presents the visualization results of **CoupledNet** on three representative partial differential equations (PDEs) from the Jaxpi benchmark, including Advection, Allen–Cahn, and Burgers equations. Each figure shows the predicted solution, analytical solution, and absolute error.

## K. Sectral Bias experiments on 1D Possion equation

To analyze the spectral bias in PINN training, we follow the experimental protocol of Wang et al. (2021b). We conduct experiments on the 1D Poisson equation to examine the evolution of the neural tangent kernel (NTK) spectrum during training. All models—including MLP, PirateNet, and CoupledNet—use two hidden layers with a width of 64 and the same activation function. No additional PINN training techniques (e.g., NTK reweighting, adaptive loss scaling, or residual regularization) are applied to ensure a fair comparison. The NTK matrix is computed at initialization and periodically during training to track the eigenvalues and the corresponding eigenvectors. The frequency spectrum of each eigenvector is visualized as a heatmap to illustrate how different architectures capture frequency components during training.



*Figure 14.* Evolution of frequency components corresponding to the top-3 NTK eigenvectors of CoupledNet.

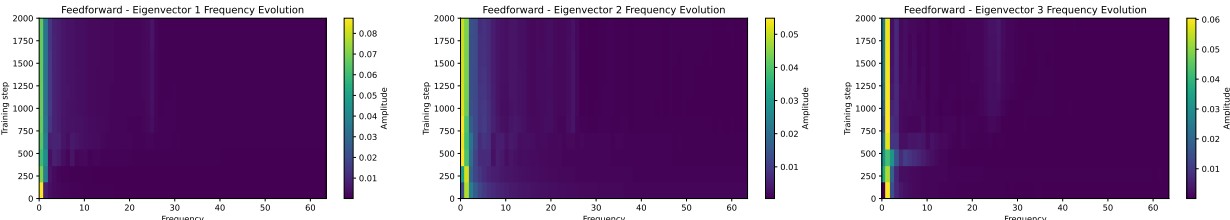

*Figure 15.* Evolution of frequency components corresponding to the top-3 NTK eigenvectors of MLP.



*Figure 16.* Evolution of frequency components corresponding to the top-3 NTK eigenvectors of PirateNet.

## L. Verification of the Summation Invariance in Bias-Free Layer Normalization

To verify the numerical stability and structural constraint of CoupledNet, we track the summation of the outputs from the bias-free LayerNorm layers throughout the training process. As shown in Figure 17, the accumulated sum remains bounded within $1.75 \times 10^{-6}$, which can be attributed solely to floating-point precision errors. This result confirms that CoupledNet strictly preserves the determinant constraint of the Jacobian matrix, i.e., $\det(\mathbf{J}) = 1$, throughout training.

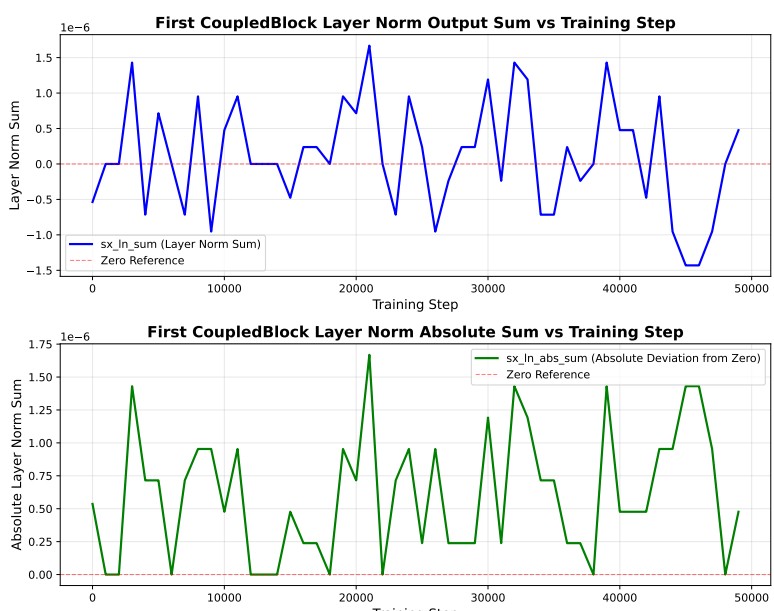

*Figure 17.* Tracking of the bias-free LayerNorm output summation during CoupledNet training. The values remain bounded within $1.75 \times 10^{-6}$ due to floating-point errors, confirming that CoupledNet rigorously satisfies the Jacobian determinant constraint $\det(\mathbf{J}) = 1$.

