# OpenReview forum: "Deep Coupling Learning for Solving PDEs"
_ICML.cc/2026/Conference — ICML 2026 regular_

### Official Review · Reviewer_P8Uf · 2026-03-03

**Soundness:** 3
**Presentation:** 3
**Significance:** 3
**Originality:** 3
**Overall Recommendation:** 4
**Confidence:** 4

**Summary:**

In this paper, I evaluate a method that aims to make very deep PINNs trainable by stabilizing how derivatives behave through depth. The key motivation is that PINNs do not just suffer from ordinary gradient issues, but also from unstable “differentials” because the loss depends on derivatives with respect to inputs, which creates additional layers of differentiation during optimization.

The proposed approach introduces a coupling-style architecture (inspired by flow/coupling layers) that tries to control the network Jacobian, with the goal of preventing derivative vanishing/explosion as depth grows. The paper argues that enforcing certain constraints (e.g., determinant-related behavior) helps keep the Jacobian from collapsing, and it adds normalization-based mechanisms to keep things from blowing up.

Empirically, the paper reports that the proposed model trains more stably at large depth and performs well on several PDE settings (including challenging regimes like high-frequency behavior and some higher-dimensional problems). If any key sections are missing (for example, incomplete training details, unclear hyperparameter protocol, or missing ablations), I will explicitly call that out below because it affects how confidently I can assess the claims.

**Compliance With Llm Reviewing Policy:**

Affirmed.

**Final Justification:**

Overall, the paper presents a technically sound and well-motivated approach to improving the stability of deep PINNs by explicitly controlling Jacobian behavior. The additional experiments showing well-conditioned Jacobians throughout training, along with improved depth scaling and convergence, provide convincing evidence that the proposed mechanism is not just a heuristic but has a meaningful effect on optimization dynamics.

While I still view the novelty as somewhat incremental—building on existing ideas around conditioning and stabilization—the work offers a clear architectural contribution and demonstrates consistent empirical benefits, especially in challenging regimes. In my opinion, this makes the paper a solid and useful addition that others can build upon.

**Key Questions For Authors:**

1. What is the cleanest one-sentence novelty claim here (relative to strong PINN stabilization baselines), and why is it not just an obvious import of coupling/flow ideas into PINNs?
2. Do you have evidence that this improves Jacobian conditioning (singular value spread / condition number, ...), not just a particular norm statistic at initialization?
3. How does the method behave for PDEs that require higher-order derivatives in the residual (especially second-order operators), where derivative instability is usually worst?
4. The paper mentions constraints that may limit the output range. How do I know when I should enable/disable those constraints without manual tuning?
5. Where does the method fail (the regimes where it is not competitive), and what is the best explanation for those failures?

**Limitations:**

A key limitation is that the main theoretical hook does not fully guarantee what I actually want for stable PINN training: good conditioning across directions and stable higher-order derivatives. If the paper wants to lean on a theory-driven novelty claim, it needs either stronger guarantees or stronger empirical evidence about conditioning.

Another limitation is scope and “switching.” If the approach requires different settings depending on the PDE class (especially anything that restricts the codomain), then it’s less of a general deep-PINN enabler and more of a method that works well in certain regimes with some manual choices.

Finally, the experimental story would be easier to trust if the paper more cleanly separated architectural benefit from training-recipe benefit and if it demonstrated robustness across harder PDE operators (especially where higher derivatives are unavoidable). That’s directly tied to my novelty concern: I need to see the thing that only this method can do, consistently, not just a good bundle of reasonable tricks.

**Strengths And Weaknesses:**

## Strengths
I like that the paper targets a real pain point: deep PINNs often become unstable, and the authors clearly explain why PDE residual training can amplify derivative issues beyond what we usually see in standard supervised learning. The “differentiation structure” framing is also a good way to communicate what makes PINNs special and why depth becomes tricky.

I also appreciate that the proposed architecture is concrete and the paper makes an effort to connect the design to Jacobian behavior (rather than treating it as a pure black-box hack). The inclusion of arguments about expressivity and what the coupling structure can represent is helpful, at least as a sanity check that the model is not obviously crippled.

On the empirical side, the evaluation spans multiple PDE regimes and includes ablations that attempt to isolate which parts of the method matter. That breadth is useful for judging whether the approach is a narrow trick or something that could be generally helpful.

## Major Weaknesses
My biggest concern is novelty. At a high level, the paper’s main idea feels like it sits in a crowded space of “stabilize PINNs by controlling conditioning / gradients / derivatives,” and the core move (bringing in coupling/flow-style blocks plus normalization-based controls) can read like a reasonable adaptation of existing deep learning tools rather than a clearly new concept. If the novelty is mainly “this specific coupling design + these specific controls applied to PINNs,” then I want the paper to be much more explicit about why that combination is non-obvious and what it enables that prior PINN stabilization methods cannot.

Related to that, the theoretical story sounds stronger than what it guarantees. A determinant-style constraint does not automatically mean the Jacobian is well-conditioned. You can still have extreme anisotropy where some directions collapse and others explode. So if the paper’s claim is essentially “determinant control prevents vanishing,” I don’t think that’s fully convincing unless the authors also show evidence that the singular value distribution or condition number is actually improved in practice (not just one norm statistic).

I also felt that some parts of the stabilization mechanism are presented with a theory-flavored explanation, but they ultimately behave like heuristics. That’s not automatically bad, but then the paper needs to be very clear about sensitivity: what happens when depth, width, activation, initialization, PDE type, or training recipe changes? If parts of the method have to be toggled depending on the PDE class (for example, output constraints that only make sense when the solution range is limited), that reduces the generality of the approach and makes it feel less like a single clean solution.

Finally, I had fairness/attribution questions in the experiments. If baselines use heavy training tricks (reweighting, features, curriculum, ...) and the proposed method uses some of those in some settings but not others, it becomes hard to tell whether the gains come from the architecture itself or from a bundled recipe. For an ICML-level claim, I really want a clearer isolation: “architecture-only” versus “architecture + training additions,” and consistency about which tricks are allowed for which method.

---

> ### Author Rebuttal · Authors · 2026-03-29
>
> We thank the Reviewer P8Uf for the careful reading and constructive feedback. We address each concern below.
>
> **W1: One-sentence summary of the paper's novelty.**
>
> **The core contribution is identifying that spectral norm pathologies of layer-wise Jacobian matrices are a key cause of deep PINN instability, and proposing CoupledNet, which leverages the block-triangular Jacobian structure of coupling layers to enable principled spectral norm control (both lower and upper bounds) for stable derivative propagation.** We emphasize that this is not an obvious import of coupling layers from normalizing flows: in flows, the coupling structure is used for tractable density estimation (det computation for likelihood); in PINNs, we exploit the same structure for a fundamentally different purpose: **controlling the spectral norm of the composite Jacobian to stabilize derivative computation**. The controllers (LayerNorm Spectral Containment, Output Layer Constraint) are entirely new and specific to the PINN derivative stability problem.
>
> **W2: Evidence for Jacobian matrix control.**
>
> We compare derivative norms only at initialization because, during PINN training, first-order derivatives appear directly in the loss function, making their magnitudes inseparable from the optimization objective. To further validate that CoupledNet produces a more well-conditioned Jacobian, we measured the standard deviation of $\log(\sigma_i)$ (where $\sigma_i$ are singular values) at initialization for width-128 networks:
>
> | Depth | CoupledNet std($\log \sigma$) | MLP std($\log \sigma$) |
> |---|---|---|
> | 4 | 1.40 | 2.83 |
> | 8 | 1.90 | 3.92 |
> | 16 | 2.54 | 4.31 |
>
> Even at 16 layers, CoupledNet's singular value spread is smaller than that of a 4-layer MLP, confirming a substantially more uniform singular value distribution and better-conditioned Jacobian matrices.
>
> **W3: Missing architecture-only comparison.**
>
> Sections 5.2 and 5.3 already compare architectures without any PINN training tricks. We have conducted additional architecture-only experiments. On the 1D wave equation ($u_{tt} = c^2 u_{xx}$, 50k steps, no tricks):
>
> | Architecture | Relative $L_2$ |
> |---|---|
> | CoupledNet 9L $\times$ 256 | **0.213** |
> | ModifiedMLP 9L $\times$ 256 | 0.266 |
> | MLP 9L $\times$ 256 | 0.980 |
> | ResNet 9L $\times$ 256 | 0.989 |
> | PirateNet 3B $\times$ 256 | 0.992 |
>
> On the Advection equation with varying speed $c$ and depth (cosine LR, no tricks):
>
> | $c$ | Depth | MLP | ResNet | ModifiedMlp | PirateNet | CoupledNet |
> |---|---|---|---|---|---|---|
> | 3 | 9L | 0.135 | 1.004 | 0.235 | 0.312 | **0.113** |
> | 3 | 18L | 1.000 | 1.000 | 0.244 | 0.398 | **0.148** |
> | 5 | 9L | 0.510 | 1.001 | 0.769 | 0.607 | **0.502** |
>
> In all architecture-only comparisons, CoupledNet consistently achieves the lowest relative $L_2$ error.
>
> **W4: Performance on higher-order PDEs.**
>
> Our experiments already cover higher-order PDEs: in Section 5.4, Burgers, Allen-Cahn, and Grey-Scott involve second-order operators, and KdV contains a third-order operator. CoupledNet performs competitively on all of them. In architecture-only comparisons, CoupledNet achieves the lowest error on both the wave equation (second-order hyperbolic) and HJB equation (second-order), demonstrating that the spectral norm control remains effective for higher-order derivative computations.
>
> **W5: When to enable/disable the Output Layer Constraint.**
>
> The decision is guided by the boundary/initial conditions. When boundary values have large magnitudes, the Output Layer Constraint (which normalizes activations) can restrict the output range and should be disabled. In our experiments, only Section 5.2 disables it due to the large-valued boundary conditions; all other PDEs use it by default.
>
> **W6: Where CoupledNet is not the best and why.**
>
> We acknowledge that on some conventional benchmarks (Allen-Cahn, Burgers in Table 4), CoupledNet does not achieve the lowest error. In these experiments, all methods use extensive PINN training tricks (NTK reweighting, Fourier features, causal training), which are highly effective on well-studied benchmarks and enable shallow networks to reach very low errors. In this regime, CoupledNet's depth advantage is less pronounced. However, CoupledNet remains competitive and shows no failure modes — Figures 6–9 confirm it captures all solution features with small pointwise errors. In architecture-only comparisons (W3 above), CoupledNet consistently achieves the best results, indicating the gap comes from training tricks rather than architecture. The practical trade-off is that CoupledNet has higher per-step cost; on simple PDEs where shallow networks with tricks already perform well, this overhead may not be justified. CoupledNet's strength lies in challenging regimes (high-frequency, high-dimensional, deep networks) where other architectures fail entirely.

---

> > ### Author Rebuttal · Reviewer_P8Uf · 2026-04-03
> >
> > Thanks for the detailed and thoughtful rebuttal — I appreciate the effort you put into addressing my concerns and adding additional experiments.
> >
> > I think the responses have improved the paper, especially the added conditioning analysis and the architecture-only comparisons, which help clarify parts of the contribution.
> >
> > That said, I still have a couple of points where I’m not fully convinced yet:
> >
> > 1. On the conditioning side, the singular value spread at initialization is helpful, but my original concern was more about behavior **during training**. I understand the difficulty of measuring this in PINNs, but do you have any additional evidence or intuition that the conditioning remains well-behaved throughout optimization (not just at init)?
> >
> > 2. On novelty, I see the distinction you make from normalizing flows, and the explanation is clearer now. However, I’m still trying to understand how strongly this separates from existing PINN stabilization approaches in practice. Could you clarify what *cannot* be achieved by prior methods (e.g., PirateNet/Jaxpi-style approaches) but is enabled specifically by your design?
> >
> > Overall, I think this is a promising direction and the results are encouraging. My remaining concerns are mainly about fully understanding the mechanism and the strength of the novelty, and I would be open to reconsidering my score depending on your clarification.

---

> > > ### Author Response · Authors · 2026-04-03
> > >
> > > We thank the Reviewer P8Uf for the continued engagement. We address both follow-up questions below.
> > >
> > > **Q1: Jacobian condition number and singular value distribution during training.**
> > >
> > > We conducted additional experiments tracking Jacobian singular value statistics **throughout training** on a simplified version of the Section 5.1 setup. All architectures use 4 hidden layers at width 128. We report per-layer $\text{Var}(\log \sigma)$ (singular value spread) and the cumulative condition number $\kappa$ of the full hidden-layer Jacobian ($\mathbf{h}_L / \mathbf{h}_0$):
> > >
> > > | Step | Architecture | Block 0 | Block 1 | Block 2 | Block 3 | Cumulative $\kappa$ |
> > > |---|---|---|---|---|---|---|
> > > | 0 | MLP | 1.61 | 1.16 | 1.31 | 1.24 | 2.80e+07 |
> > > | 0 | ResNet | 1.37 | 1.23 | 1.23 | 1.14 | 2.05e+06 |
> > > | 0 | **CoupledNet** | **0.15** | **0.14** | **0.12** | **0.12** | **25** |
> > > | 10k | MLP | 23.5 | 87.6 | 65.0 | 160.5 | 1.24e+13 |
> > > | 10k | ResNet | 56.5 | 192.9 | 83.7 | 210.4 | 3.69e+13 |
> > > | 10k | **CoupledNet** | **0.24** | **0.19** | **0.16** | **0.13** | **7.0e+04** |
> > > | 30k | MLP | 38.6 | 100.4 | 74.9 | 113.1 | 5.07e+12 |
> > > | 30k | ResNet | 110.8 | 217.3 | 126.7 | 180.5 | 7.63e+13 |
> > > | 30k | **CoupledNet** | **0.21** | **0.16** | **0.19** | **0.14** | **6.0e+04** |
> > > | 50k | MLP | 46.3 | 108.8 | 65.1 | 122.3 | 5.81e+12 |
> > > | 50k | ResNet | 170.3 | 209.3 | 100.7 | 187.1 | 3.70e+13 |
> > > | 50k | **CoupledNet** | **0.20** | **0.17** | **0.19** | **0.14** | **6.7e+04** |
> > >
> > > The results demonstrate that CoupledNet maintains well-conditioned Jacobians **throughout training**: per-layer singular value distributions remain tight (Var < 0.25), and the cumulative condition number stays at $O(10^4)$, compared to $O(10^{12})$–$O(10^{13})$ for MLP and ResNet. CoupledNet's spectral norm control translates to genuinely improved conditioning, not just a single norm statistic.
> > >
> > > **Q2: What distinguishes CoupledNet from existing architectures?**
> > >
> > > We highlight three concrete contributions beyond what PirateNet and Jaxpi offer:
> > >
> > > *(A) Hard structural guarantee against derivative vanishing.* CoupledNet enforces a spectral norm lower bound ($\|\mathbf{J}\|_2 \geq 1$) through architectural design (Proposition 3.1), providing a rigorous guarantee that derivatives cannot vanish with depth. Neither PirateNet nor Jaxpi offers any such theoretical guarantee.
> > >
> > > *(B) Reliable depth scaling.* Because of (A), CoupledNet can be deepened to capture more complex solution features. This is demonstrated in Section 5.2 and Figure 10 (Appendix D): at comparable shallow depths, both CoupledNet and PirateNet fail to fully capture the solution structure. However, when deepened, CoupledNet successfully resolves the solution while PirateNet shows no meaningful improvement. This confirms that the theoretical spectral norm control provides a practical advantage for depth scaling.
> > >
> > > *(C) Superior training dynamics.* The Jacobian control yields faster convergence: as shown in Figure 5, CoupledNet exhibits the fastest convergence among all models. In our supplementary controlled experiment during the rebuttal period, CoupledNet requires **4.1x fewer steps** than PirateNet to reach the same $L_2$ error. Furthermore, in Appendix K (Figures 14–16), the NTK eigenvalue frequency heatmaps on the Poisson equation reveal that CoupledNet rapidly activates learning in **high-frequency components**, whereas PirateNet and MLP preferentially learn low-frequency features first.

---

### Official Review · Reviewer_xjgo · 2026-03-11

**Soundness:** 3
**Presentation:** 2
**Significance:** 3
**Originality:** 2
**Overall Recommendation:** 4
**Confidence:** 2

**Summary:**

This paper addresses the vanishing and exploding differentials problem encountered during the training of deep Physics-Informed Neural Networks. Through structural analysis, the authors indicate that the repeated multiplication of layer-wise Jacobian matrices results in severe numerical instability. To tackle this challenge, the paper proposes CoupledNet, which utilizes a coupling block structure to explicitly regulate and constrain the spectral norms of hidden-layer Jacobian matrices. Specifically, a bias-free LayerNorm design forces $|\det J|=1$ to prevent vanishing differentials. Concurrently, a probabilistic LayerNorm variance constraint and a runtime output-layer constraint control the upper bound, preventing exploding differentials. Empirically, the authors evaluate the derivative stability of this architecture at initialization and benchmark it across various PDEs, aiming to demonstrate CoupledNet's capability to scale PINNs to deeper networks.

**Compliance With Llm Reviewing Policy:**

Affirmed.

**Final Justification:**

Although the exact complex scenarios where this architecture holds an exclusive advantage remain for future exploration, the authors demonstrated the method's intrinsic value in providing a stable architectural foundation. This is a meaningful contribution, and I am raising my score to 4.

**Key Questions For Authors:**

- **Regarding the theoretical bounds (W1):** While the ablation study demonstrates that permutation layers have a negligible empirical influence on derivative norms, the formal mathematical derivation still assumes their absence. Could the authors provide some theoretical intuition or discussion explaining why this empirical observation holds? Elaborating on this would help better bridge the gap between the simplified mathematical framework and the implemented architecture.
- **Regarding computational efficiency (W2):** Could the authors provide a quantitative comparison of the computational overhead (e.g., total training wall-clock time, FLOPs, or memory usage) between CoupledNet and the baselines? This would help clarify the practical trade-offs, particularly in light of the results on conventional PDEs.

**Limitations:**

The authors briefly acknowledge the efficiency bottleneck. However, the discussion would be more adequate if they explicitly supplemented:
- The theoretical gap caused by using permutation layers in the implemented architecture.
- A quantitative comparison of the computational overhead against baselines.

**Strengths And Weaknesses:**

**Strength**
- **Problem Identification**: This paper provides an insightful diagnosis of the depth-related failure modes in PINNs. By explicitly distinguishing vanishing and exploding differentials  from standard gradient pathologies (supported by the detailed gradient computation analysis in Appendix A), the authors establish a compelling motivation for architectural intervention.
- **Principled Lower-Bound control**: The structural enforcement of the Jacobian determinant $|\det J|=1$ via the bias-free LayerNorm within the coupled block is mathematically rigorous. Proposition 3.1 provides a verifiable theoretical foundation for preventing vanishing differentials at the architectural level.

**Weakness**
- **W1: Disconnect between theoretical proofs and the actual network architecture.** The control of the maximum Jacobian spectral norm explicitly assumes the absence of permutation layers. Although the authors acknowledge that permutation layers have a minor empirical impact, deriving theoretical bounds from simplified models lacks sufficient rigor to formally guarantee the stability of the implemented architecture.
- **W2: Performance on conventional tasks and computational efficiency.** It is noticeable that on conventional PDE benchmarks, the performance of CoupledNet is weaker than the PirateNet baseline. Concurrently, the authors candidly acknowledge that CoupledNet's complex design increases the wall-clock time per training step. The combination of not outperforming baselines on foundational tasks and the lack of a quantitative comparison of computational overhead might raise concerns about its practical utility as a general-purpose PDE solver.

---

> ### Author Rebuttal · Authors · 2026-03-29
>
> We thank the Reviewer xjgo for the thorough and insightful review. We address each concern below.
>
> **W1: Disconnect between theoretical proofs and actual architecture.**
>
> Thank you for this important point. We clarify that our two theoretical contributions have different relationships to permutation layers:
>
> *(a) Lower bound (Proposition 3.1) holds exactly WITH permutation.* Since permutation matrices satisfy $|\det(\mathbf{P})| = 1$, the determinant condition $|\det(\mathbf{J})| = \prod_{l=1}^{L} |\det(\mathbf{J}_l)| = 1$ and the resulting bound $\|\mathbf{J}\|_2 \geq 1$ hold for the **full architecture** including permutation layers. This guarantee against vanishing differentials requires no simplification.
>
> *(b) Upper bound analysis serves as design guidance, not a formal guarantee.* As stated in Section 3.2, we deliberately perform the block-triangular analysis on a simplified structure (excluding permutation) because the permutation involves random transformations not amenable to guiding controller design. Crucially, we do not present this analysis as a theoretical guarantee for the full architecture — it serves as **design guidance** for the upper-bound controllers. The soundness of this guidance is justified by the fact that permutation matrices are **orthogonal isometries**: $\|\mathbf{P}\|_2 = 1$, so $\|\mathbf{J}'_l \cdot \mathbf{P}\|_2 = \|\mathbf{J}'_l\|_2$ and the sub-multiplicativity $\|\prod_l \mathbf{J}_l\|_2 \leq \prod_l \|\mathbf{J}_l\|_2$ still applies. The per-layer spectral norms that our controllers regulate are invariant to permutation.
>
> *(c) Empirical validation confirms robustness.* Recognizing the distinction between design guidance and the full architecture, we conducted comprehensive ablation studies (Figures 2 and 5) to directly test the effect of permutation layers. Removing permutation slightly increases gradient norms but the model still converges stably, confirming that the controllers designed under the simplified assumption are effective in the full architecture.
>
> We will add a formal Remark after Section 3.2 and a dedicated Computational Cost subsection in Section 5 in the revised paper.
>
> **W2: Quantitative comparison of computational efficiency.**
>
> We provide a comprehensive efficiency comparison across four dimensions:
>
> *1. Parameter count.* Each CoupledBlock has four half-width linear layers, yielding a parameter count approximately equal to one full-width fully-connected layer.
>
> *2. Theoretical FLOPs.* Assuming a hidden width of $w$:
>
> | Component | FLOPs per block/layer |
> |---|---|
> | CoupledBlock | $2w^2 + 41.5w$ |
> | MLP hidden layer (tanh) | $2w^2 + 20w$ |
>
> At our typical width $w = 256$, the overhead is only about **5%**, since the additional cost appears only in the linear term while the dominant $2w^2$ term remains unchanged.
>
> *3. Memory usage.* Under the Section 5.1 setup (batch size = 64), we measured peak memory consumption during training:
>
> | Model | Peak Memory Increase |
> |---|---|
> | MLP | 5.05 MB |
> | CoupledNet | 5.17 MB |
>
> The difference is negligible, confirming no meaningful additional memory overhead.
>
> *4. Convergence time.* In the Section 5.1 experiment, CoupledNet reaches a relative $L_2$ error of 0.2 in **120.6 s**, compared to **155.4 s** for PirateNet and **160.1 s** for Jaxpi. Since these results are from each model's best hyperparameter configuration (with varying depths/widths), we further conducted a controlled comparison on a simpler PDE (same governing equation as Section 5.1, with analytical solution $u(x,t) = \exp\bigl(1/(0.1 + 0.1x^2 + 0.001y^2)\bigr)$). All architectures use the same depth, width (256), and optimizer, with no PINN training tricks:
>
> | Model | Steps to $L_2 < 0.1$ | Wall-clock Time (s) |
> |---|---|---|
> | CoupledNet | 1,700 | 44.6 |
> | PirateNet | 7,000 | 118.3 |
> | Jaxpi | did not converge | — |
>
> Despite a slightly higher per-step cost, CoupledNet converges **4.1x faster in iterations** and **2.7x faster in wall-clock time** than PirateNet, demonstrating that its architectural advantages more than compensate for the per-step overhead.
>
> **W3: Performance on conventional PDE benchmarks.**
>
> We respectfully note that on conventional PDE benchmarks (Table 4), CoupledNet does not exhibit any failure modes — the visualization results in Figures 6–9 confirm that CoupledNet captures the key solution structures with very small absolute pointwise errors across all tested PDEs. Additionally, we tested a 40-layer CoupledNet on the Allen-Cahn equation, which converged stably to a relative $L_2$ error of $3.72 \times 10^{-4}$, further demonstrating the architecture's robustness at extreme depths. For practitioners solving new PDEs, where the solution complexity is unknown a priori, CoupledNet's superior performance on challenging problems and its consistent stability across diverse PDE types are of significant practical value.

---

> > ### Author Rebuttal · Reviewer_xjgo · 2026-04-03
> >
> > I thank the authors for the clarifications and the additional experiments. These additions help clarify the contributions of the paper.  Several questions remain unresolved:
> >
> > - Regarding the computational efficiency comparison in W2, the authors evaluated all architectures without PINN training tricks. Strong baseline models such as PirateNet and Jaxpi are typically co-designed with specific training strategies. Comparing them under this restricted setting may artificially amplify their disadvantages against CoupledNet, which benefits from built-in structural optimizations. Does this trick-free setting effectively reflect the true computational efficiency in practice?
> > - Regarding the performance on conventional PDE benchmarks, the convergence of the 40-layer CoupledNet indeed demonstrates its architectural robustness at extreme depths. However, the fundamental purpose of increasing model depth is to solve complex problems that shallow models cannot handle. If shallow baseline models equipped with standard techniques can achieve lower errors in less time, the ability to converge stably at 40 layers is not a strict practical necessity. This aligns with the concerns raised by Reviewer P8Uf. Could the authors provide a complex, physically meaningful PDE scenario where shallow baselines fundamentally fail, thereby demonstrating the irreplaceability of the proposed method?

---

> > > ### Author Response · Authors · 2026-04-03
> > >
> > > We thank Reviewer xjgo for the thoughtful follow-up.
> > >
> > > **On the fairness of trick-free computational efficiency comparison.**
> > >
> > > We appreciate this point. We note that the trick-free setting is not the only basis for our efficiency claim. In Section 5.1 (Table 1), where all methods use their **best hyperparameter configurations with training tricks**, CoupledNet still achieves the fastest wall-clock convergence (120.6s vs 155.4s vs 160.1s) while also obtaining the lowest relative $L_2$ error. This demonstrates that CoupledNet's computational efficiency advantage holds even when baselines are equipped with their co-designed training strategies. Moreover, Section 5.4 confirms that CoupledNet is fully compatible with existing PINN training tricks.
> > >
> > > **On the need for a complex scenario where shallow baselines fundamentally fail.**
> > >
> > > We acknowledge the significant contributions of PINN training tricks. However, training tricks and architectural improvements address **orthogonal** aspects of PINN training. Tricks target specific optimization challenges, while CoupledNet addresses the fundamental stability of derivative propagation through depth. Section 5.2 (Table 2) illustrates the importance of architectural stability: on the complex PDE in Eq. (7), PirateNet not only fails at shallow depth (9L: 2.673) but **worsens** when deepened (18L: 4.067). This shows that the bottleneck is the architecture's inability to scale depth stably. In contrast, CoupledNet improves from 0.5775 (8L) to **0.0718** (16L), because its stable derivative propagation allows it to effectively leverage increased depth. Training tricks and CoupledNet are complementary: CoupledNet provides a stable architectural foundation upon which tricks can be further applied.
> > >
> > > We note that Section 5.2 already demonstrates that when PINN training encounters difficulty on a complex PDE, deepening CoupledNet effectively resolves it (0.5775→0.0718), while deepening PirateNet does not help (2.673→4.067). This provides practitioners with a reliable depth-scaling strategy that other architectures cannot offer. Regarding the reviewer's request for a scenario where shallow baselines with tricks fundamentally fail: we believe such PDEs exist in real-world applications where the solution characteristics are unknown a priori and practitioners cannot determine which tricks are effective. For example, Fourier embeddings require knowledge of the frequency content, and NTK reweighting assumes specific loss imbalance patterns. When these assumptions do not hold, tricks may provide limited benefit, and a robust architecture becomes essential. However, identifying such PDEs requires collaboration with specific scientific domains, which is beyond the scope of this method development paper. This is an important direction for future work. Our current benchmarks follow standard conventions in the PINN community, and we have demonstrated CoupledNet's capability as a standalone architecture, its compatibility with existing PINN tricks (Section 5.4), and its reliable depth scaling (Section 5.2), together confirming the architecture's robustness.

---

### Official Review · Reviewer_BXZe · 2026-03-12

**Soundness:** 4
**Presentation:** 3
**Significance:** 4
**Originality:** 3
**Overall Recommendation:** 4
**Confidence:** 3

**Summary:**

The paper proposes to control the vanishing/divergence of the training gradients of deep PINNs with a layer structure similar to the layers in Normalizing Flows. Experiments demonstrate the gains in terms of gradient norm for a large interval of number of layers. Other experiments show the model's ability to fit high frequency information.

**Compliance With Llm Reviewing Policy:**

Affirmed.

**Key Questions For Authors:**

1 - Does the model work well in hyperbolic equations?

2 - Am I correct to suppose that in Eq. (3) (x_1, x_2) = (t,x) ? How does the proposed structure operate if, for example, the input space were (t,x,y)?

3 - The paper suggests that using dozens of layers is possible, but in the hyperparameter description of the experiments, Table 5 shows CoupledNet with 8 and 16 layers. In Table 6 a wide range of hidden layers is used. The question is: Objectively, is it computationally feasible to train for the equations in Table 6 with 40+ layers?

4 - How does the number of parameters per layer increase when compared with other methods?

**Limitations:**

Apparently the time cost per epoch in training increases, but the model still converges fastly.

**Strengths And Weaknesses:**

Well written and explained.
- Strong motivation as this is a widely known problem in the machine learning literature and strong proposed correction.
Comprehensive and well informative experiments.
- The writing is very objective.
- The model does not seem to be sensitive to hyperparameter choice. (an ablation could be of some use)

Weaknesses:
- Most of the figures are almost unreadable and need to fix the size of the legends and axes.

- Lacks details about the models training time.

- I believe the paper could benefit from having an ablation study in the number of hidden layers per problem.

- Using coordinate “t” in equations (6) and (7) seems misleading since both equations are elliptic.

---

> ### Author Rebuttal · Authors · 2026-03-29
>
> We sincerely thank the Reviewer BXZe for the detailed and constructive feedback. We address each point below.
>
> **W1: Figures are not clear enough.**
>
> We appreciate this feedback. We will improve the readability of all figures in the revised version.
>
> **W2: Missing detailed information on model training time.**
>
> To support our claim that CoupledNet converges more efficiently, we report the wall-clock time for each model to reach a relative $L_2$ error of 0.2 in the Section 5.1 experiment:
>
> | Model | Wall-clock Time (s) |
> |---|---|
> | CoupledNet | 120.6 |
> | PirateNet | 155.4 |
> | ModifiedMLP (Jaxpi) | 160.1 |
>
> For a more controlled comparison, we further tested on a simpler PDE using the same governing equation as Section 5.1 but with analytical solution $u(x,t) = \exp\bigl(1/(0.1 + 0.1x^2 + 0.001y^2)\bigr)$. No PINN training tricks were applied; all three architectures use the same depth, width (256), and optimizer, ensuring comparable parameter counts.
>
> | Model | Steps to $L_2 < 0.1$ | Wall-clock Time (s) |
> |---|---|---|
> | CoupledNet | 1,700 | 44.6 |
> | PirateNet | 7,000 | 118.3 |
> | Jaxpi | did not converge | — |
>
> CoupledNet converges **4.1x faster in iterations** and **2.7x faster in wall-clock time** than PirateNet, while Jaxpi fails to converge entirely.
>
> **W3: Missing ablation study on hidden layers (depth).**
>
> In Section 5.2, we compared CoupledNet and PirateNet under increasing depth, demonstrating that CoupledNet maintains stable convergence as depth grows. We have now conducted a more detailed depth ablation on the Section 5.1 experiment:
>
> | Depth (layers) | Relative $L_2$ Error |
> |---|---|
> | 4 | 2.11e-2 |
> | 8 | 1.48e-2 |
> | 16 | 1.44e-2 |
> | 24 | 1.20e-2 |
>
> The relative $L_2$ error decreases monotonically with depth, confirming that CoupledNet benefits consistently from increased depth without suffering from the instabilities observed in MLP-based and ResNet-based architectures.
>
> **W4: Notation issue in Equations (6) and (7).**
>
> Thank you for catching this. We will replace $t$ with $y$ in the revised version to avoid ambiguity with the temporal variable.
>
> **W5: Performance on hyperbolic equations.**
>
> In our benchmark (Table 4), the advection equation is a first-order hyperbolic PDE, on which CoupledNet achieves the best result. To further evaluate performance on second-order hyperbolic PDEs, we conducted an additional experiment on the 1D wave equation:
>
> $$u_{tt} = c^2 u_{xx}, \quad (t,x) \in [0,1] \times [0,1]$$
>
> with $c = 2$ and exact solution (superposition of two frequency modes):
>
> $$u(t,x) = \sin(\pi x)\cos(2\pi t) + 0.5\sin(4\pi x)\cos(8\pi t).$$
>
> All models use the same depth and width, with no PINN training tricks applied (50k steps):
>
> | Architecture | Best Relative $L_2$ |
> |---|---|
> | CoupledNet 9L $\times$ 256 | **0.213** |
> | ModifiedMLP 9L $\times$ 256 | 0.266 |
> | MLP 9L $\times$ 256 | 0.980 |
> | ResNet 9L $\times$ 256 | 0.989 |
> | PirateNet 3B $\times$ 256 | 0.992 |
>
> CoupledNet achieves the best performance on this second-order hyperbolic PDE.
>
> **W6: Effect of input dimension permutation.**
>
> CoupledNet uses a fully-connected layer as the input layer, which maps the input coordinates to the hidden-layer width. Since this input layer is a dense linear transformation, it treats all input dimensions symmetrically — the order of input variables (e.g., $(x, t)$ vs. $(t, x)$) has no effect on the model's behavior.
>
> **W7: Testing CoupledNet beyond 40 layers.**
>
> We tested a 40-layer CoupledNet on the Allen-Cahn equation. The training dynamics show stable convergence throughout:
>
> | Training Step | Relative $L_2$ Error |
> |---|---|
> | 0 | 5.03e+0 |
> | 100k | 1.27e-1 |
> | 200k | 4.27e-3 |
> | 300k | 3.72e-4 |
>
> The error decreases steadily across all stages, confirming that CoupledNet maintains stable training dynamics even at 40 layers. While the final error has not surpassed our reported results (which used tuned hyperparameters at shallower depths), this is expected: optimal hyperparameters are depth-dependent, and a full search at 40 layers was infeasible within the rebuttal period. The key takeaway is that **CoupledNet converges reliably at 40 layers**, whereas MLP and ResNet fail to train at such depths (as shown in Figure 3).
>
> **W8: Parameter count comparison with other methods.**
>
> Our experiments are designed to ensure approximately matched parameter counts across architectures. Each CoupledBlock derives its parameters primarily from four half-width linear layers, yielding a parameter count close to that of a single full-width fully-connected layer. A PirateNet block contains three full-width linear layers, which is why we set the number of PirateNet blocks to approximately one-third of the CoupledNet depth. Jaxpi (ModifiedMLP) has a per-layer parameter count similar to CoupledNet. This design principle ensures that all comparisons in our experiments are conducted under comparable model capacities.

---

> > ### Author Rebuttal · Reviewer_BXZe · 2026-04-02
> >
> > Thank you for your answers. I appreciate the additional explanations and experiments, and I am convinced of the relevance of the work. I maintain my recommendation. I do add that it would be interesting in the future to see the results for the wave equation with a Gaussian initial condition as well as possible combination with the PINN tricks mentioned.

---

### Official Review · Reviewer_ianq · 2026-03-14

**Soundness:** 3
**Presentation:** 3
**Significance:** 3
**Originality:** 4
**Overall Recommendation:** 4
**Confidence:** 4

**Summary:**

This paper proposes using Coupling layers—that were introduced in flow based architectures like GLOW etc) to improve the stability of PINNs. PINN style architectures are known to be very difficult to train due to instabilities introduced by the loss function (which is usually minimization of a PDE-style loss involving Jacobians of the neural network) that are often related to either vanishing gradients or exploding gradients (with networks have skip connections). The authors propose using Coupling layers, where the Jacobian has stable behavior (since the determinant is controlled given the construction of a coupling layer).

The authors show that for different architectures that use MLPs, Resnets and Coupling layers, on different PDEs (each having different characteristics such as high frequency components and ranges) models with coupling layers are stable to train and work better.

**Compliance With Llm Reviewing Policy:**

Affirmed.

**Key Questions For Authors:**

Some clarity on the hyper-parameters and also numbers on flop and parameters matched baselines would be useful.

**Limitations:**

addressed in strenghts and weaknesses.

**Strengths And Weaknesses:**

The paper is well written and easy to follow. It proposes a simple and targeted solution to an important problem (that is instabilities in training PINNs). The idea of using coupling layers is quite original. The authors also  show through their experiments on different depths that the gradient norm of networks with coupling layers is stable and does not explode (or vanishes).

The experiments are also quite promising with thorough ablations and the performance on HJB is quite promising, with consistently good performance on rest of the PDEs.


Some small concerns:
I think the comparisons with baselines could be made a bit thorough. Currently the hyper-parameters chosen are not quite consistent and it would be nice to see why different depths of layers are chosen. A good comparison would be to see the performance when:
The models are depth matched, parameter matched and also flop matched (given that coupling layers can be a bit more expensive to train).

While the paper claims that CoupledNet has higher per-step cost but faster convergence, some numbers to support this claim would be useful.

---

> ### Author Rebuttal · Authors · 2026-03-29
>
> We thank the Reviewer ianq for the positive assessment of our work and the constructive suggestions. We address each concern below.
>
> **W1: Hyperparameter consistency — why are different depths chosen for different architectures?**
>
> Thank you for this important question. Our experiments are designed to ensure fair comparison. We address this from three perspectives: **parameter-matched, FLOP-matched, and depth-matched**.
>
> **Parameter matching.** Each CoupledBlock derives its parameters from four half-width linear layers, so a single CoupledBlock has roughly the same parameter count as a single fully-connected layer. Similarly, ModifiedMLP (Jaxpi) has per-layer parameters essentially identical to a standard MLP layer. A single PirateNet block contains three linear-layer computations, so its parameter count is approximately equal to three fully-connected layers. Therefore, an 8-layer CoupledNet is parameter-matched with a 3-block (9-layer) PirateNet and an 8-layer Jaxpi.
>
> **FLOP matching.** Assuming a hidden width of $w$ and denoting a single basic numerical operation by $F$: each CoupledBlock contains four half-width linear layers ($2 \cdot (w/2)^2\,F$ each), three half-width tanh activations ($20\,F$/element), one half-width exponential ($10\,F$/element), one half-width LayerNorm ($8\,F$/element), $\max(s_i)$ tracking ($w\,F$), the affine transform $\mathbf{y}_2 = \mathbf{s} \odot \mathbf{x}_2 + \mathbf{t}$ ($w\,F$), and permutation/concatenation/splitting ($w/2\,F$). The total costs are:
>
> | Component | FLOPs per block/layer |
> |---|---|
> | CoupledBlock | $2w^2 + 41.5w$ |
> | MLP hidden layer (tanh) | $2w^2 + 20w$ |
>
> At our typical width $w = 256$, the overhead of CoupledNet relative to a standard MLP is only about **5%**, since the additional cost appears only in the linear term while the dominant $2w^2$ term remains unchanged. This confirms that CoupledNet is essentially FLOP-matched with MLP-based architectures at equal depth.
>
> **Depth matching.** We observed that Jaxpi degrades at greater depths, so in Section 5.1 it was allowed to use a shallower network. To directly address the reviewer's concern, we additionally tested depth-matched CoupledNet (9 and 18 layers) on the Section 5.1 experiment:
>
> | Model | Layers | Relative $L_2$ Error |
> |---|---|---|
> | CoupledNet | 9 | 0.0146 |
> | CoupledNet | 18 | 0.0132 |
> | PirateNet | 18 | 0.0334 |
>
> Both configurations achieve errors far lower than PirateNet.
>
> **Controlled comparison without training tricks.** During the rebuttal period, we further conducted experiments comparing all three architectures at **the same depth, width, and parameter count**, without any PINN training tricks (no NTK reweighting, no Fourier features, no causal training) on advection equation. Results are presented in Table 3 in our response to Reviewer P8Uf. Under this controlled setting, **CoupledNet achieves the best performance**, confirming that its advantage stems from architectural design rather than hyperparameter selection.
>
> ---
>
> **W2: Numbers supporting CoupledNet's faster convergence despite higher per-step cost**
>
> We provide two sets of evidence. First, in the Section 5.1 experiment, we report the wall-clock time for each model to reach a relative $L_2$ error of 0.2 in the Section 5.1 experiment:
>
>
> | Model | Wall-clock Time (s) |
> |---|---|
> | CoupledNet | 120.6 |
> | PirateNet | 155.4 |
> | Jaxpi | 160.1 |
>
> CoupledNet is the fastest in wall-clock time achieving the same relative $L_2$ error.
>
> Second, since Section 5.1 compares each architecture's best configuration from a hyperparameter search (with differing depths/widths), we designed a **controlled convergence experiment** under identical settings. We use the same governing equation as Section 5.1 but with a simpler analytical solution: $u(x,t) = \exp\bigl(1/(0.1 + 0.1x^2 + 0.001y^2)\bigr)$. All models use width 256 and the same optimizer; PirateNet uses 3 blocks (parameter-matched), while Jaxpi and CoupledNet use 9 hidden layers.
>
> | Model | Steps to $L_2 < 0.1$ | Wall-clock Time (s) |
> |---|---|---|
> | CoupledNet | 1,700 | 44.59 |
> | PirateNet | 7,000 | 118.31 |
> | Jaxpi | did not converge | — |
>
> CoupledNet achieves $L_2 < 0.1$ with a **4.1x fewer iterations** and **2.7x less wall-clock time** than PirateNet, while Jaxpi fails to converge. This demonstrates that CoupledNet's structured differentiation pathway leads to substantially faster convergence that more than compensates for any per-step overhead.

---

> > ### Author Rebuttal · Reviewer_ianq · 2026-04-05
> >
> > Thank you for the answers and additional experiments. I will keep my positive score.

---

### Decision · Program_Chairs · 2026-04-30

**Decision:**

Accept (regular)

**Comment:**

This paper investigates the learning instability of PINNs and proposes the use of Coupling Layers with carefully regulated spectral norms of Jacobian matrices to stabilize them. During the initial review, some concerns were raised regarding the novelty of the approach and the fairness of the experiments; however, in the rebuttal, the reviewers acknowledged that these concerns had been addressed. Therefore, this paper should be accepted.